# Visual Attention Never Fades: Selective Progressive Attention ReCalibration for Detailed Image Captioning in Multimodal Large Language Models

**Mingi Jung** [1]   **Saehyung Lee** [1]   **Eunji Kim** [1]   **Sungroh Yoon** [1 2 3]

## Abstract

Detailed image captioning is essential for tasks like data generation and aiding visually impaired individuals. High-quality captions require a balance between precision and recall, which remains challenging for current multimodal large language models (MLLMs). In this work, we hypothesize that this limitation stems from weakening and increasingly noisy visual attention as responses lengthen. To address this issue, we propose SPARC (Selective Progressive Attention ReCalibration), a training-free method that enhances the contribution of visual tokens during decoding. SPARC is founded on three key observations: (1) increasing the influence of all visual tokens reduces recall; thus, SPARC selectively amplifies visual tokens; (2) as captions lengthen, visual attention becomes noisier, so SPARC identifies critical visual tokens by leveraging attention differences across time steps; (3) as visual attention gradually weakens, SPARC reinforces it to preserve its influence. Our experiments, incorporating both automated and human evaluations, demonstrate that existing methods improve the precision of MLLMs at the cost of recall. In contrast, our proposed method enhances both precision and recall with minimal computational overhead. code: https://github.com/mingi000508/SPARC

## 1. Introduction

Multimodal Large Language Models (MLLMs) have recently gained traction as a transformative approach in artificial intelligence by integrating visual and linguistic modal-

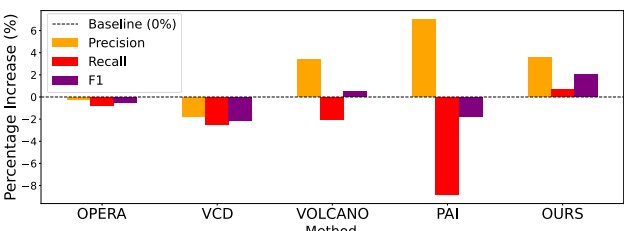

*Figure 1.* Percentage change in precision, recall, and F1-score compared to the results before applying each method.

ities (Li et al., 2023a; Liu et al., 2024c; Lin et al., 2024). These models leverage the powerful language capabilities of Large Language Models (LLMs) to generate textual descriptions from visual inputs (Bai et al., 2023; Touvron et al., 2023; Abdin et al., 2024; Peng et al., 2023). This capability enables MLLMs to effectively perform a wide range of tasks, including Visual Question Answering (VQA), multimodal reasoning, and image captioning (Chen et al., 2024; Wang et al., 2024; Li et al., 2024a; Liu et al., 2024e).

Among these tasks, detailed image captioning aims to produce comprehensive yet accurate textual descriptions that capture both key elements and subtle nuances of an image. This capability is particularly important in applications such as content creation (Belyaeva et al., 2023; Han et al., 2023) or assistive technology for visually impaired individuals (Hao et al., 2024). However, a major challenge in current approaches is hallucination—where models introduce incorrect or irrelevant details—compromising the reliability of the generated captions and limiting their practical utility (Bai et al., 2024; Liu et al., 2024d; Cui et al., 2023).

Although many methods have been proposed to reduce hallucinations (Huang et al., 2024; Leng et al., 2024; Lee et al., 2023; Liu et al., 2025), we demonstrate that existing methods primarily enhance *precision*—the extent to which a caption accurately reflects an image—at the expense of *recall*—the extent to which a caption comprehensively describes the image—often resulting in captions that, while more precise, omit essential details. To highlight this trade-off, we use the CHAIR benchmark (Rohrbach et al., 2018) to assess the effectiveness of existing hallucination mitigation methods in

[1]Department of Electrical and Computer Engineering, Seoul National University, Seoul, South Korea [2]Interdisciplinary Program in Artificial Intelligence, Seoul National University [3]AIIS, ASRI, INMC, and ISRC, Seoul National University. Correspondence to: Sungroh Yoon <sryoon@snu.ac.kr>.

*Proceedings of the 42nd International Conference on Machine Learning*, Vancouver, Canada. PMLR 267, 2025. Copyright 2025 by the author(s).

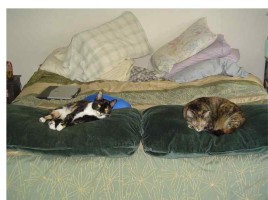

**Prompt**: *Please describe this image in detail.*

**Generated Text**

Baseline

The image features a bed with two `cats` lying on it. One cat is located on the left side of the bed, while the other cat is on the right side. Both cats are resting comfortably on pillows, with one cat lying on a pillow near the center of the bed and the other cat lying on a pillow closer to the right side. In addition to the `cats`, there are two laptops on the bed. One laptop is placed on the left side of the bed, and the other laptop is on the right side. A remote control can also be seen on the left side of the bed.

Ours

The image features a bed with two `cats` lying on top of it. One cat is on the left side of the bed, while the other cat is on the right side. Both cats are sleeping comfortably, with one of them wearing a blue pillow on its head. In addition to the `cats` there are several pillows on the bed, including a laptop computer placed on the left side of the bed. The bed is covered with a green blanket, and the colorful pillows add a cozy and inviting atmosphere to the bed.

**Attention**

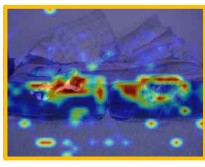

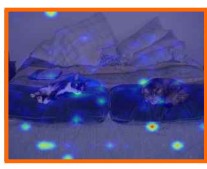

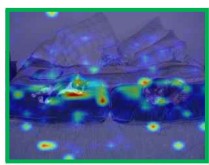

*Figure 2.* Visualization of image token attention at different context lengths. As the generation context length increases, image attention diminishes, reducing reliance on visual inputs. Our approach mitigates this by preserving image attention, reducing hallucinations and enabling more detailed captions.

terms of recall and precision. Notably, our analysis reveals a previously overlooked limitation: these methods significantly reduce model recall (Figure 1). Our work aims to introduce a new approach that balances precision and recall.

In this work, we hypothesize that MLLMs do not fully realize their potential in terms of precision and recall. We attribute this limitation to the model's focus on visual tokens gradually weakening as it generates longer text and its increasing sensitivity to irrelevant noise, as illustrated in Figure 2. To address these issues, we propose **SPARC** (Selective Progressive Attention ReCalibration), a training-free method designed to enhance the contribution of key visual tokens during the decoding process of MLLMs. Specifically, SPARC is built upon the following three principles: (1) naively increasing the influence of all visual tokens reduces recall; therefore, SPARC selectively amplifies the influence of visual tokens; (2) despite noisy visual attention patterns, SPARC identifies key visual tokens by leveraging attention differences across time steps; (3) to compensate for the weakening influence of visual tokens, SPARC accumulates reinforcement effects to sustain their impact

Our experiments demonstrate that SPARC significantly im-

proves caption quality, outperforming existing methods. Unlike conventional methods that struggle to balance precision and recall, SPARC effectively enhances both. Furthermore, human evaluations validate the superiority of SPARC in producing more precise and comprehensive captions.

Our contributions are summarized as follows:

- We empirically show that existing MLLM hallucination mitigation methods overlook recall.

- We propose SPARC, a novel attention-based method that improves MLLM image captioning in both precision and recall. We provide empirical evidence supporting the design choices of the proposed method.

- Through automated and human evaluations, we show that SPARC, while training-free and computationally efficient, effectively enhances both precision and recall.

## 2. Related Work

### 2.1. Mitigating Hallucinations in MLLMs

MLLMs often generate hallucinations, where text is inconsistent with visual input, and numerous studies have aimed to address this issue through various methods (Li et al., 2023b; Liu et al., 2024d; Gunjal et al., 2024). Decoding-based methods tackle hallucination by penalizing uncertain token generations through techniques like text aggregation analysis (Huang et al., 2024), corrupted visual inputs (Leng et al., 2024; Gong et al., 2024). Self-refinement strategies are also employed to iteratively align generated captions with visual content (Zhou et al., 2023; Lee et al., 2023). In addition, research indicates that hallucinations often arise from an over-reliance on textual context while neglecting visual information (Zhu et al., 2024; Liu et al., 2025). To address this imbalance, decoding strategies and attention calibration techniques have been developed to enhance the utilization of relevant visual elements based on their importance (Huo et al., 2024; Liu et al., 2025; Li et al., 2025). Hallucination issues intensify with long-form text generation, as models rely more on text and less on image content (Favero et al., 2024; Lee et al., 2024; Zhong et al., 2024). Methods such as shortening text length (Yue et al., 2024), segmenting refinement (Lee et al., 2024), and leveraging image-only logits and contrastive decoding (Zhong et al., 2024; Favero et al., 2024) have been explored. Existing solutions have not effectively addressed the challenge of enhancing context-relevant visual information, as vision-related signals tend to weaken over longer contexts.

### 2.2. Visual Attention in MLLMs

MLLMs utilize transformer-based architectures with attention mechanisms to integrate visual and textual modali-

ties (Vaswani, 2017; Basu et al., 2024; Osman et al., 2023). During text generation, attention weights do not always focus on the most relevant visual tokens (Zhang et al., 2024; Woo et al., 2024; Jiang et al., 2024). Prior studies have shown that enhancing image attention can help mitigate hallucinations by improving the model's ability to better align with relevant visual content (Li et al., 2024b; Xing et al., 2024). Efforts to achieve this include increasing image attention, adjusting positional embeddings (Xing et al., 2024; Li et al., 2025), and correcting biases that direct attention to irrelevant regions. (Jiang et al., 2024; Gong et al., 2024; Kang et al., 2025). Several approaches have been proposed, such as boosting overall image attention (Liu et al., 2025; Jiang et al., 2024) and reweighting key tokens (Xing et al., 2024) to better focus on meaningful visual regions. Our findings show that as the model generates longer text, its focus on key visual tokens weakens, while sensitivity to irrelevant noise increases. Existing methods struggle to retain key visual tokens in such cases, making it difficult to preserve their relevance. In contrast, our approach reinforces key visual tokens during decoding, ensuring their continued importance. This improves caption detail and accuracy with minimal computational cost.

## 3. Why More Attention Doesn't Always Help

Intuitively, enhancing the influence of visual tokens during MLLM decoding could lead to higher-quality captions. A recent study (Liu et al., 2025) has shown that amplifying visual attention can indeed improve MLLMs' precision. However, Figure 1 reveals that this improvement comes at the cost of a significant decline in recall. In this section, we explore this issue in depth, analyzing why the naive amplification of visual attention—increasing all attention weights assigned to visual tokens—can lead to captions with lower recall. We identify key factors contributing to these limitations and discuss potential strategies to mitigate them.

### 3.1. Does More Attention Reduce Diversity?

When generating a detailed and comprehensive caption for an image, a person's gaze naturally moves across different regions of the image. Similarly, we hypothesize that MLLMs producing high-recall captions attend to diverse locations within the image during decoding. Based on this hypothesis, we analyze the impact of naive attention amplification on visual attention dynamics. Specifically, we examine the pairwise distances between visual attention patterns obtained during decoding. If an MLLM shifts its attention across various regions of an image while generating a caption, the distances between its visual attention patterns should be large. To investigate this, we generate 3,000 captions using LLaVA-1.5 (Liu et al., 2024a) and a subset of DOCCI (Onoe et al., 2025). During generation, we store the

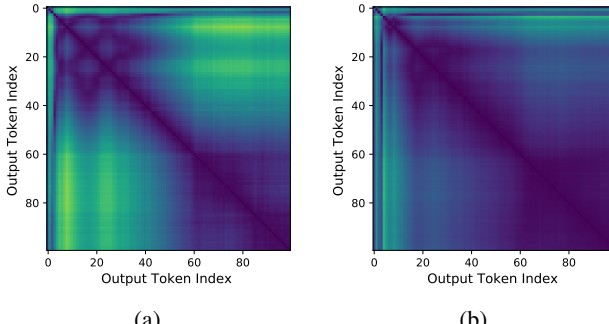

*Figure 3.* Visual attention diversity comparison between (a) the baseline model and (b) the naive attention enhancement approach. The naive approach reduces visual attention diversity, indicating ineffective adaptation to important visual tokens

normalized visual attention weights for the first 100 tokens of each caption. We then compute a pairwise distance matrix ($100 \times 100$) of visual attention patterns. The distance is calculated using the Wasserstein distance (Vallender, 1974; Shen et al., 2018).

Figure 3 illustrates visual attention diversity during captioning, comparing the baseline model with a naive attention enhancement method (Liu et al., 2025). The results show that naive attention enhancement (right) yields lower distance values compared to the baseline (left), indicating a reduction in visual attention diversity. Notably, naive attention enhancement causes the model's visual attention pattern to remain static throughout caption generation. Consequently, we consider this a key factor contributing to recall degradation, as it reduces the diversity of objects mentioned in the captions and limits their descriptiveness.

These findings highlight the need for a carefully designed approach to visual attention enhancement. Properly identifying and emphasizing the most relevant visual tokens is crucial for generating captions that are both contextually rich and informative.

### 3.2. Longer Context, More Noise?

We analyzed how the model's attention to visual tokens evolves throughout the caption generation process to investigate why directly amplifying attention based on its magnitude can reduce attention diversity. For this analysis, we extracted attention weights from one of the middle-to-late transformer layers, averaged them across attention heads, and normalized the values across all visual tokens. Figure 4 provides a visualization that reveals several notable patterns.

In the early stages of caption generation, attention is primarily directed toward image regions corresponding to salient visual elements, thereby ensuring that the visual attention

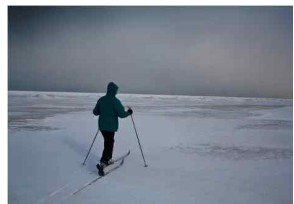

**Prompt:** *Please describe this image in detail.*

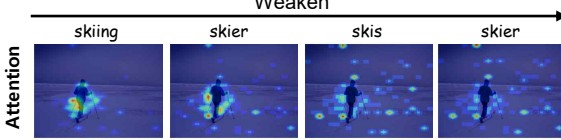

The image features a person wearing a green jacket and __skiing__ across a snow-covered field. The __skier__ is holding ski poles in their hands, and they appear to be enjoying the winter activity. The person is the main focus of the scene, with the __skis__ clearly visible beneath them. The snowy landscape extends beyond the skier, creating a serene and picturesque winter scene. The __skier__'s presence in the middle of the field adds a sense of adventure and excitement to the image.

*Figure 4.* Visualization of the temporal dynamics of image attention during caption generation. Early in the process, attention is focused on contextually relevant regions, but as the caption grows longer, it increasingly shifts toward noise or consistently high-attention tokens.

mechanism itself remains aligned with key parts of the image. However, as the caption progresses, this alignment weakens. The intensity of attention on relevant regions diminishes, while attention increasingly concentrates on noise or on specific visual tokens that consistently attract high attention weights. These noisy attention patterns are often unrelated to the context of generation and tend to recur at the same visual token positions throughout the generation process. These visual tokens, often located in background region, may capture global context (Darcet et al., 2024; Kang et al., 2025). However their excessive attention dominance can overwhelm local, task-relevant signals. Consequently, commonly used strategies that amplify tokens according to their current attention values may inadvertently exacerbate this problem: as the caption becomes longer, boosting already high-attention tokens can reinforce irrelevant or noisy regions, ultimately impairing the model's ability to focus on truly important parts of the image.

These findings underscore the need for an adaptive visual attention mechanism that can effectively identify and prioritize contextually meaningful visual tokens while filtering out noise.

### 3.3. Does Longer Context Weaken Visual Focus?

We analyzed the model's focus on visual information, considering how it changes with caption length from an attention perspective. Specifically, we generated 3,000 captions using LLaVA-1.5 with the DOCCI subset and then analyze

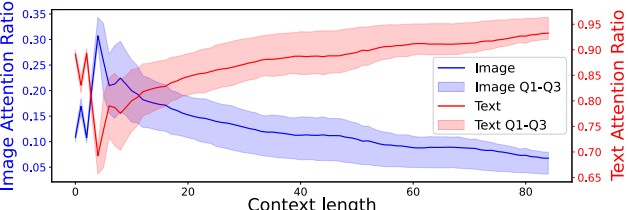

*Figure 5.* Average attention weight trends for text and image tokens as a function of context length during caption generation. As the context length increases, the proportion of attention allocated to image tokens gradually decreases compared to text tokens. This indicates a significant shift in focus from visual to textual elements during the later stages of caption generation.

the total attention weight allocated to image tokens and text tokens for each output token during the caption generation process and plot their distribution with respect to the context length. Figure 5 presents the results of this analysis, illustrating that as the context length increases, the proportion of attention allocated to image tokens gradually decreases compared to text tokens. This finding aligns with previous studies (Favero et al., 2024; Lee et al., 2024), which reported that longer captions often lead to increased hallucinations due to a decline in the model's focus on relevant visual information.

To address the diminishing focus on visual information, which can weaken the image's influence and potentially lead to hallucination, it is essential to ensure sustained visual attention throughout the caption generation process. A gradual increase in the emphasis on visual tokens, particularly in the later stages, helps counteract this declining trend. By progressively reinforcing the model's visual focus over time, it can better retain and utilize the image context, ultimately enhancing the accuracy and relevance of the generated captions.

The detailed implementation details of the above experiments can be found in Appendix B.1 .

## 4. Method

In Section 3, we observed that as the context length generated by the MLLM increases, the attention to visual tokens becomes noisier and its proportion decreases. To address this issue, we propose Selective Progressive Attention Re-Calibration (**SPARC**), a training-free approach that incurs minimal additional computational cost. SPARC consists of two key mechanisms: 1) **Token selection using the Relative Activation Score**, which robustly selects visual tokens relevant to the actual context and resists the increasing noise as the context length grows, and 2) **Selective Progressive Attention Re-Calibration**, which progressively reinforces the attention to important visual tokens during each genera-

tion step, thereby alleviating the decline in visual attention as the context length increases.

### 4.1. Preliminaries: Attention Mechanisms in MLLMs

MLLMs are designed to process both image and text inputs, generating text outputs in an autoregressive manner (Li et al., 2024a; Lin et al., 2024). When generating the $i$-th text token, the models attend to an input sequence of length

$$N_i = N_{\text{image}} + N_{\text{inst}} + (i - 1),$$

consisting of $N_{\text{image}}$ image embeddings, $N_{\text{inst}}$ instruction tokens, and the $i - 1$ tokens generated so far.

At the $l$-th layer and $h$-th attention head, let the query, key, and value vectors be $Q_i^{(l,h)}$, $K_j^{(l,h)}$, $V_j^{(l,h)} \in \mathbb{R}^d$, where $i$ indicates the current token being generated and $j$ ranges over all $N_i$ tokens in the input (*i.e.*, $j \in \{1, 2, \ldots, N_i\}$). The attention weight $\alpha_{i,j}^{(l,h)}$, denoting how much the $i$-th token attends to the $j$-th, is given by

$$\alpha_{i,j}^{(l,h)} = \text{softmax}_j\big(A_{i,j}^{(l,h)}\big), \quad A_{i,j}^{(l,h)} = \frac{\big(Q_i^{(l,h)}\big)^\top K_j^{(l,h)}}{\sqrt{d}}. \tag{1}$$

Here, $d$ is the scaling factor. Using these attention weights, the output representation $o_i^{(l,h)}$ for the $i$-th token at the $l$-th layer and $h$-th attention head is

$$o_i^{(l,h)} = \sum_{j=1}^{N_i} \alpha_{i,j}^{(l,h)} V_j^{(l,h)}. \tag{2}$$

The outputs from all $H$ attention heads at the $l$-th layer are concatenated and projected back to the original dimension.

A key challenge in MLLMs is their limited ability to effectively utilize image information during text generation (Zhu et al., 2024; Zhong et al., 2024). To address this issue, several methods have been proposed to explicitly enhance attention to image tokens during the text generation process (Jiang et al., 2024; Zhang et al., 2024). A simple way to boost attention to image tokens is to modify as follows (Liu et al., 2025):

$$A_{i,j}^{(l,h)} \leftarrow A_{i,j}^{(l,h)} + \alpha \cdot |A_{i,j}^{(l,h)}|, \tag{3}$$

where $\alpha$ is a scaling factor that amplifies the attention given to image tokens.

These approaches leverage the premise that attention mechanisms capture critical visual information, reinforcing attention to important visual tokens in proportion to their significance.

### 4.2. Token Selection: Relative Activation Score

To effectively identify contextually relevant visual tokens as the generated context expands, we propose a **Relative activation score**, a dynamic metric that prioritizes tokens with significant relative increases in attention. This score is designed to compare the current attention value with a smoothed historical trend, emphasizing relative changes rather than absolute magnitudes. By normalizing these changes with respect to the smoothed values, our method ensures that meaningful variations remain detectable, even as attention scales evolve.

Instead of relying solely on instantaneous attention values, our approach focuses on temporal variations in attention. By comparing the current attention value against its historical trend, we minimize the influence of static biases or localized noise, effectively highlighting tokens with sharp increases in relevance—even if their raw attention values are not the highest. To stabilize the process, we apply an Exponential Moving Average (EMA), which smooths out transient fluctuations and prevents brief spikes or dips in attention from disproportionately influencing the selection process. As the context length grows, attention values may diminish overall, which can obscure important variations. By dynamically adapting to these scale changes through normalization, our method remains sensitive to significant shifts in attention, regardless of the overall reduction in attention magnitude over time.

Concretely, we employ an Exponential Moving Average (EMA) to track the historical trend of attention weights, allowing us to dynamically compare the current attention weight to its smoothed past values. This enables our method to detect relative increases in attention with high precision, while filtering out transient noise. we first maintain an EMA of the attention weights up to the $(i - 1)$-th step:

$$\tilde{a}_{i-1,j}^l = \beta \tilde{a}_{i-2,j}^l + (1 - \beta) a_{i-1,j}^l, \tag{4}$$

where $\beta \in [0, 1]$ isis the smoothing factor that determines the relative weighting of past and current attention values. $a_{i-1,j}^l$ represents the attention weights for the $j$-th token at the $(i - 1)$-th step, averaged across all attention heads $h$.

Using this smoothed trend, we define the **Relative Activation Score** for the $j$-th image token as:

$$r_{i,j}^l = \frac{a_{i,j}^l - \tilde{a}_{i-1,j}^l}{\tilde{a}_{i-1,j}^l}, \quad \text{for } j \in \{1, 2, \ldots, N_{\text{image}}\}. \tag{5}$$

Here, $\tilde{a}_{i-1,j}^l$ represents the EMA-smoothed attention weight for token $j$ at layer $l$. This scoring mechanism emphasizes tokens with substantial relative increases in attention, effectively prioritizing those that become more salient over time while minimizing the impact of static or irrelevant tokens.

Based on these relative activation scores, we apply a thresholding mechanism to select the most relevant image tokens. The set of selected tokens $S_i$ is defined as:

$$S_i = \{j \mid r_{i,j}^l > \tau\}, \tag{6}$$

where $\tau$ is a predefined threshold. Tokens exceeding this threshold are treated as the significant visual elements for generating the $i$-th text token. By dynamically adjusting to shifts in attention patterns, this method focuses on visually relevant tokens while mitigating noise and static biases.

### 4.3. Selective Progressive Attention Re-Calibration

As text generation progresses, attention to important image regions often diminishes, leading to a misalignment between visual and textual contexts. To address this issue, we propose **Selective Progressive Attention Re-Calibration (SPARC)**, a mechanism that dynamically reinforces attention on relevant image tokens at each step of text generation. SPARC ensures that contextually significant visual tokens maintain their importance throughout the captioning process, enabling the model to produce richer and more accurate descriptions.

The core idea of SPARC is to compute a cumulative relevance measure for each image token and adjust attention weights dynamically based on this measure. To achieve this, we introduce the **Selection Count**, which quantifies the evolving importance of each image token during text generation. The Selection Count is formally defined as:

$$c_{i,j} = \sum_{k=1}^{i-1} \mathbf{1}(j \in S_k), \tag{7}$$

where $\mathbf{1}(\cdot)$ is an indicator function that returns 1 if $j$ is selected at step $k$, and 0 otherwise. This metric accumulates the number of times each token has been deemed relevant in prior steps, reflecting its sustained importance in the evolving context.

At each text generation step $i$, we first identify the set of contextually relevant tokens $S_i$ using our **Token Selection** method and update the Selection Count $c_{i,j}$ for each token $j$. Based on the updated count, we recalibrate the attention weights by amplifying those of frequently selected tokens:

$$a_{i,j}^{l} \leftarrow a_{i,j}^{l} \cdot \alpha^{c_{i,j}}, \tag{8}$$

where $\alpha > 1$ is a scaling parameter controlling the amplification of tokens with higher cumulative relevance. This exponential scaling prioritizes consistently relevant tokens while maintaining adaptability to new contexts, mitigating the decline in attention weights observed in longer text generation.

To further enhance computational efficiency, we implement this recalibration by directly adjusting the token's value vectors. Specifically, for each $j \in S_i$, we update:

$$V_j^{(l,h)} \leftarrow V_j^{(l,h)} \cdot \alpha. \tag{9}$$

This is possible because key-value caching in large language models stores value vectors for each token (Wan et al., 2023).

Leveraging this cached information, SPARC efficiently updates value vectors without additional memory overhead, ensuring seamless recalibration through weighted sums of value vectors (e.g., Equation (2)).

In summary, SPARC addresses attention decay by progressively reinforcing the significance of contextually relevant image tokens. This method prevents attention sinks and noise while preserving alignment between visual and textual contexts, resulting in more accurate, diverse, and visually grounded captions.

## 5. Experiments

We evaluate captioning quality by comparing baseline model performance using existing approaches versus our method. We also assess models with and without the proposed method, conduct human evaluations against conventional techniques, and analyze the methods introduced in Section 4. Qualitative results are provided in Appendix A.

### 5.1. Experimental Setup

**Models**  We conduct experiments on three widely used multi-modal language models (MLLMs): LLaVA-1.5 (Liu et al., 2024a), LLaVA-Next (Liu et al., 2024b), and Qwen2-VL (Wang et al., 2024), each with 7B parameters. For each model, we generat captions for given images using the prompt: "Please describe this image in detail." The maximum token length for caption generation is set to 512 across all models.

**Metrics**  We use two evaluation metrics in our experiments. The first metric, CLAIR (Chan et al., 2023), measures overall caption quality by assessing alignment with reference captions. It determines whether the generated and reference captions effectively describe the same image, with higher scores indicating better quality. CLAIR leverages GPT-4o (Hurst et al., 2024) for reliable evaluation of detailed and accurate captions. The second metric, CHAIR (Rohrbach et al., 2018), assesses hallucination by comparing objects mentioned in generated captions with those in reference captions. It provides precision-recall metrics to evaluate the trade-off between correctly identified and erroneously included objects.

**Datasets**  For CLAIR evaluation, we use the IIW-400 (Garg et al., 2024) and DOCCI (Onoe et al., 2025) datasets. IIW-400 consists of 400 image-caption pairs, while DOCCI contains 15K pairs. Both datasets provide highly detailed, hallucination-free captions, making them well-suited for evaluating caption quality. For CHAIR evaluation, we utilize the MS-COCO 2014 validation dataset (Lin et al., 2014). This dataset includes ground-truth object annotations, which enable the calculation of CHAIR metrics by compar-

*Table 1.* Comparison of CLAIR scores on the IIW-400 and DOCCI datasets for the baseline model with existing methods and our proposed approach. The results highlight the performance improvements achieved by our method.

| METHOD | IIW-400 | DOCCI |
|---|---|---|
| BASELINE | 56.36 | 59.26 |
| OPERA | 51.02 (-5.34) | 56.81 (-2.45) |
| VCD | 52.16 (-4.20) | 55.60 (-3.66) |
| VOLCANO | 55.84 (-0.52) | 61.09 (+1.83) |
| PAI | 56.86 (+0.50) | 60.09 (+0.83) |
| **OURS** | **61.49** (+5.13) | **62.70** (+3.44) |

ing the objects mentioned in the generated captions with the reference objects.

**Implementation Details**   To compare our method with existing approaches, we conduct experiments on the baseline model, LLaVA-1.5. The hyperparameters for existing methods are implemented as specified in prior research. For our method, the following parameters are applied across all models: the scaling factor $\alpha$ is set to 1.1, the smoothing factor $\beta$ is set to 0.1, and the selection threshold $\tau$ is adjusted for each model. Specifically, $\tau$ is set to 1.5 for LLaVA-1.5, 4.0 for LLaVA-Next, and 3.0 for Qwen2-VL. For token selection, we extract visual attention from layer 20 for LLaVA-1.5 and LLaVA-Next, while layer 18 is used for Qwen2-VL. On the other hand, attention recalibration is applied across all layers' attention.

## 5.2. Results

**Comparison with Existing Approaches**   We compar our method with the existing approaches on LLaVA-1.5, using CLAIR scores on the IIW-400 and DOCCI datasets. Table 1 summarizes the results, showing that our method achieves the highest scores across both datasets, significantly outperforming prior approaches. This improvement demonstrates that our method enhances caption generation by producing captions that are more detailed and better aligned with reference captions.

**Precision-Recall Tradeoff Analysis**   Generating high-quality captions requires a balanced improvement in both precision and recall. In the CHAIR metric, precision measures the proportion of objects in generated captions that do not appear in the reference captions, indicating hallucination. Recall quantifies the proportion of reference objects correctly identified in the generated captions. The F1 score combines these metrics to provide a holistic assessment of the trade-off between precision and recall.

We compar our method against existing approaches in terms of precision, recall, and F1 score using CHAIR, as shown

*Table 2.* Performance of various methods on detailed image captioning using the CHAIR benchmark. The table reports precision, recall, and F1-score for objects in generated captions. The best scores are **bolded**, while the second-best scores are underlined.

| METHODS | PRECISION | RECALL | F1 |
|---|---|---|---|
| BASELINE | 84.70 | 79.46 | 81.99 |
| OPERA | 84.54 | 78.82 | 81.58 |
| VCD | 83.22 | 77.50 | 80.26 |
| VOLCANO | 87.64 | 77.82 | 82.39 |
| PAI | **90.64** | 72.44 | 80.52 |
| OURS | 87.72 | **79.98** | **83.67** |

*Table 3.* Comparison of CLAIR scores on the IIW-400 dataset between the baseline and our method applied to various models. The results demonstrate a consistent performance improvement with our approach.

| MODEL | BASELINE | OURS |
|---|---|---|
| LLAVA-1.5 | 56.36 | **61.49** (+5.13) |
| LLAVA-NEXT | 58.86 | **64.94** (+6.08) |
| QWEN2-VL | 78.34 | **79.70** (+1.36) |

in Table 2. To ensure a robust evaluation, we randomly sample 500 instances and repeated the evaluation five times. OPERA (Huang et al., 2024) and VCD (Leng et al., 2024), which rely on decoding-based strategies fail to improve precision or recall. VOCANO (Lee et al., 2023), which incorporates feedback to self-revise its initial response, enhances precision but slightly reduces recall. PAI (Liu et al., 2025), which increases attention to the image while incorporating additional decoding techniques, achieves the largest precision gain but suffers the lowest recall, resulting in a lower F1 score than our method.

In contrast, our method successfully improves both precision and recall, achieving the highest F1 score. Specifically, it increases precision by 3.02%p, recall by 0.52%p, and F1 score by 1.68%p. Notably, our approach is the only one that improves recall compared to the baseline, whereas all other methods sacrifice recall to boost precision.

These results demonstrate that our method minimizes incorrect object inclusions while enhancing the model's ability to identify relevant objects. This improved balance is crucial for generating captions that are both accurate and comprehensive, setting a new benchmark for high-quality caption generation. The detailed CHAIR metric scores are provided in Appendix F.

**Performance Across Different Models**   To demonstrate the effectiveness of our method across diverse models, we evaluate it on three widely used MLLMs: LLaVA-1.5, LLaVA-Next, and Qwen2-VL. Tables 3 and 4 present the

*Table 4.* Comparison of CLAIR scores on the DOCCI dataset between the baseline and our method applied to various models. The results demonstrate a consistent performance improvement with our approach.

| MODEL | BASELINE | OURS |
|---|---|---|
| LLAVA-1.5 | 59.26 | **62.70** (+3.44) |
| LLAVA-NEXT | 62.49 | **66.99** (+4.50) |
| QWEN2-VL | 79.22 | **80.64** (+1.42) |

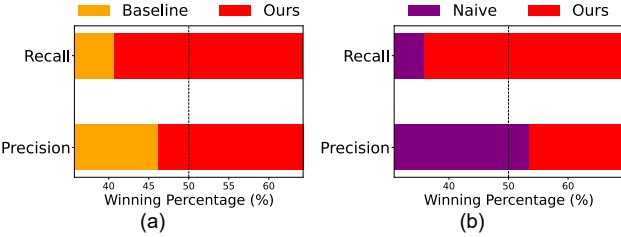

*Figure 6.* Human evaluation results showing the winning ratio (%) of our method compared to (a) the baseline and (b) naive approach in terms of precision and recall.

CLAIR scores for the IIW-400 and DOCCI datasets, respectively. For DOCCI, we randomly select 500 samples for evaluation. Across both datasets, our approach consistently improves caption quality, demonstrating its robustness across diverse model architectures. These findings confirm that our approach not only refines precision and recall but also generalizes well across different datasets and model configurations. The consistent performance gains further underscore the adaptability of our method in aligning generated captions more effectively with reference captions, as measured by the CLAIR metric.

**Human Evaluation Results**   To further assess the precision-recall tradeoff, we conduct a human evaluation. Specifically, we sample 100 captions generated by the LLaVA-1.5 model on images from the IIW-400 dataset. These captions are evaluated for precision and recall by human annotators, who compare different methods and selected the better one. The results are then aggregated into a winning ratio, showing how often our method is preferred over the baseline and naive attention enhancement approach.

As shown in Figure 6, our method achieves a higher recall compared to the baseline while also improving precision. Then compared to naive approach, our approach demonstrates superior recall with a slight decrease in precision. These results indicate that our method effectively balances recall and precision, reducing hallucinations while ensuring comprehensive caption generation.

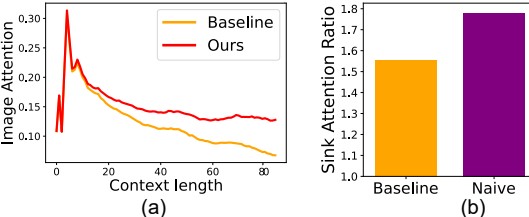

*Figure 7.* Visual Attention Analysis. (a) Change in visual attention with increasing context length. SPARC mitigates the decline compared to the baseline. (b) Ratio of attention scales between sink and non-sink visual tokens during captioning. SPARC maintains baseline-level proportions, unlike the naive approach.

### 5.3. Analyses

**Quantitative Evaluation for Visual Attention**   To analyze the change in visual token attention, we compare it against a naive approach and the baseline model. Figure 7(a) illustrates how visual attention changes with increasing context length for each method. It shows that SPARC effectively mitigates the decline in visual attention, preserving focus on visual tokens even in longer contexts.

We also quantify how much visual attention is assigned to semantically relevant image regions during the decoding process, using 5,000 randomly sampled images from the MSCOCO 2014 validation set processed by the LLaVA-1.5 model. Specifically, we calculated the total attention score allocated to image tokens within the region of that object for each generated token corresponding to a ground truth object in the caption. To identify these regions, we employed an open-vocabulary segmentation model(Ren et al., 2024; Ravi et al., 2024) to generate binary masks for all ground truth objects. During caption generation, we measured the proportion of visual attention focused on the corresponding object region relative to the total attention across all image tokens for each object-related token.

As shown in Table 5, our method assigns a higher proportion of visual attention to image-relevant regions compared to the baseline. In contrast, naive attention scaling allocates less attention to relevant image regions than the baseline. These results indicate that our method achieves better alignment between generated text and semantically relevant visual regions, suggesting that the visual attention in our model is less noisy and more accurately focused during the captioning process.

Additionally, we analyze the attention distribution over sink tokens—tokens identified as unrelated or uninformative—during the caption generation process. Following (Kang et al., 2025), which identifies sink tokens based on hidden state dimensions with exceptionally high values, we compute the ratio of attention scales between sink and non-

*Table 5.* Comparison of the proportion of visual attention focused on semantically relevant image regions between the baseline, naive attention scaling and our method.

| METHOD | ATTENTION ON RELEVANT REGIONS (%) |
|---|---|
| BASELINE | 17.85 |
| NAIVE | 15.50 (-2.35) |
| OURS | **19.17** (+1.32) |

*Table 6.* Comparison of token generation time (ms/token) between our method and existing approaches. Our method achieves efficiency comparable to the baseline, whereas other methods exhibit substantially higher computational overhead.

| MODEL | GENERATION TIME (MS/TOKEN) | $\Delta$ (%) |
|---|---|---|
| BASELINE | $30.37 \pm 0.73$ | |
| OPERA | $322.28 \pm 118.26$ | +961.3 |
| VCD | $59.44 \pm 0.84$ | +95.7 |
| VOLCANO | $109.98 \pm 17.71$ | +262.2 |
| PAI | $57.75 \pm 0.86$ | +90.2 |
| OURS | $\mathbf{31.21} \pm 0.61$ | **+2.8** |

sink tokens and plotted the results for the three approaches, as shown in Figure 7(b). The naive approach significantly increases the attention to sink tokens, which can detract from meaningful visual token focus. In contrast, SPARC maintains a sink token attention proportion comparable to the baseline, effectively avoiding unintended amplification of irrelevant tokens.

These results highlight the robustness of SPARC in preserving meaningful visual attention dynamics, even in challenging contexts with extended lengths, while avoiding the pitfalls observed in naive attention reinforcement methods. The detailed implementation details of the above experiments can be found in Appendix B.2.

**Efficiency Comparisons**    To demonstrate that our method incurs minimal computational overhead, we conducted an efficiency comparison against existing approaches. Specifically, we generated image captions using the LLaVA-1.5 model on the IIW-400 dataset with an RTX8000 GPU. For each caption, we measured the generation time per output token and then computed the average token generation time across all captions.

As shown in Table 6, our method achieves a token generation time comparable to the baseline, whereas other methods are significantly slower-by a factor of 2× to 10×. This clearly demonstrates that our approach enables efficient caption generation with minimal computational cost. Although our method incorporates attention amplification mechanisms, these introduce only negligible overhead relative to the original decoding process. In contrast, many prior methods rely on additional decoding passes, which substantially increase computational burden.

In terms of memory usage, our method only requires storing the head-wise averaged attention scores for image tokens at each layer from the previous decoding step. For instance, with LLaVA-1.5, this storage amounts to 32 layers × 576 image tokens × 2 bytes (float16), totaling less than 40 KB—an overhead that is trivial on modern hardware.

**Ablations**    Further ablation studies on the specific parameters or setting choices used in our approach are provided in Appendix C for a comprehensive analysis of their im-

pact on model performance. Briefly, we conducted ablation experiments focusing on two key design components: the token selection strategy and the progressive attention calibration mechanism. Specifically, we evaluated the performance when (1) bypassing the token selection process and applying progressive attention calibration to all image tokens, and (2) modifying the token selection strategy to use only the previous step instead of EMA. These variations highlight the individual contributions of both the token selection strategy and the progressive attention calibration in enhancing model performance.

Additionally, we conducted ablation studies on four critical parameters: the token selection layer $l$, token selection threshold $\tau$, EMA smoothing factor $\beta$, and scaling parameter $\alpha$, evaluating their impact on performance across LLaVA-1.5, LLaVA-NeXT, and Qwen2-VL. We observe consistent trends across models, with optimal performance typically achieved using mid-to-late layers for token selection and mildly scaled parameter values. While some tuning is required to achieve the best results for each model, our training-free, low computational overhead method makes hyperparameter tuning both efficient and practical, requiring minimal effort to identify optimal settings.

## 6. Conclusion

In this work, we address the challenge of balancing precision and recall in detailed image captioning for multimodal large language models. To mitigate this issue, we propose SPARC, a training-free method that enhances the contribution of visual tokens during decoding. SPARC identifies critical visual tokens by leveraging attention differences across generation steps and progressively reinforces visual attention to counteract its natural decline. Our experimental results, validated through both automated metrics and human evaluations, reveal that conventional methods often improve precision at the cost of recall. In contrast, SPARC effectively enhances both precision and recall with minimal computational overhead, offering a simple yet powerful solution for improving detailed image captioning in MLLMs.

## Acknowledgements

This work was supported by Institute of Information & Communications Technology Planning & Evaluation (IITP) grant funded by the Korea government (MSIT) [RS-2022-II220959; No.RS-2021-II211343, Artificial Intelligence Graduate School Program (Seoul National University)], the National Research Foundation of Korea (NRF) grant funded by the Korea government (MSIT) (No. 2022R1A3B1077720; 2022R1A5A708390811), the BK21 FOUR program of the Education and the Research Program for Future ICT Pioneers, Seoul National University in 2025, the Mobile eXperience(MX) Business, Samsung Electronics Co., Ltd, and a grant from Yang Young Foundation.

## Impact Statement

As AI models evolve beyond simple object recognition to generating captions that incorporate contextual and semantic understanding, the length and complexity of generated descriptions are increasing. This trend makes the challenge of balancing precision and recall in image captioning even more critical. Our research addresses this issue by improving multimodal large language models (MLLMs), which can contribute to advancements in AI for accessibility and multimodal contextual understanding. More accurate and context-aware captions can greatly benefit visually impaired individuals by providing richer and more informative descriptions. Additionally, applications in education, automated documentation, and creative content generation can be enhanced through improved captioning capabilities.

However, the societal impact of advanced image captioning must be carefully considered. While our method enhances caption quality, AI-generated descriptions can still suffer from hallucinations—incorrect details that seem plausible. As AI-generated captions become more refined and detailed, users may develop a stronger trust in their accuracy, even when the content is incorrect or misleading. This could lead to unintended consequences, such as misinterpretations of visual content or overreliance on AI-generated descriptions without verification. Ensuring that users remain aware of these limitations is crucial as AI-generated content becomes more prevalent in real-world applications.

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

# A. Qualitative Results: Enhanced Caption Quality

## A.1. LLaVA-1.5

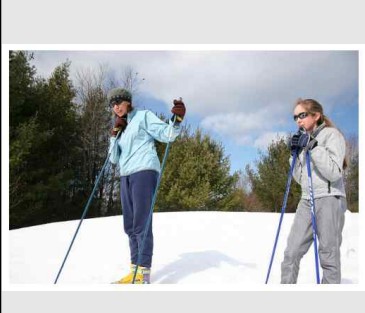

**Baseline**

The image features two women standing on a snow-covered slope, each wearing skis and holding ski poles. They are both smiling and enjoying their time on the snowy hill. The women are positioned close to each other, with one woman slightly to the left and the other slightly to the right. Their skis are visible, with one pair of skis on the left side of the image and the other pair on the right side. The women are also wearing backpacks, with one backpack on the left side and the other on the right side of the image. The scene captures a fun and active winter day spent skiing together.

**Ours**

The image features two women standing on a snow-covered slope, each wearing skis and holding ski poles. They are both smiling and enjoying their time on the snowy hill. The women are positioned close to each other, with one woman on the left and the other on the right. In addition to the skis and poles, the women are also wearing sunglasses, which can be seen on their faces. One of the women is wearing a hat, adding to the wintery atmosphere. The scene is set against a cloudy sky, creating a picturesque winter landscape.

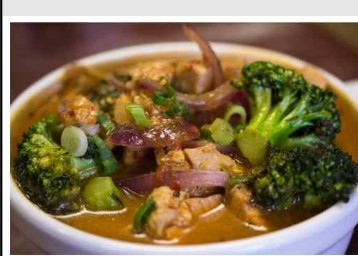

**Baseline**

The image features a bowl filled with a delicious and hearty meal, consisting of a combination of meat and broccoli. The broccoli pieces are scattered throughout the bowl, with some pieces placed closer to the top and others near the bottom. The meat, which appears to be chicken, is also present in the bowl, adding to the overall flavor and texture of the dish. The bowl is placed on a dining table, ready to be enjoyed by someone.

**Ours**

The image features a bowl filled with a delicious and hearty meal, consisting of a combination of meat and broccoli. The bowl is filled with a rich and flavorful soup, containing a generous amount of broccoli pieces. The broccoli pieces are scattered throughout the bowl, with some pieces being larger and others smaller. The soup is garnished with a variety of onions, which are also distributed throughout the bowl. The dish is served in a bowl, making it an appealing and comforting meal.

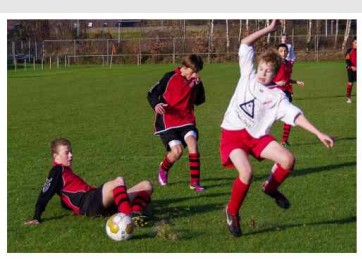

**Baseline**

The image captures a group of young boys playing soccer on a grassy field. There are at least nine players visible in the scene, with some of them actively engaged in the game, while others are standing or running around the field. A soccer ball is located in the center of the field, with one player attempting to kick it. The players are spread out across the field, with some closer to the ball and others further away. The scene is lively and full of energy, showcasing the excitement and camaraderie of the young players.

**Ours**

The image captures a group of young boys playing soccer on a grassy field. There are at least nine players visible in the scene, with some of them actively engaged in the game, while others are standing or running around the field. A soccer ball is located in the center of the field, with one boy lying on the ground, possibly after falling or being tackled. The other players are scattered across the field, with some closer to the fence and others further away

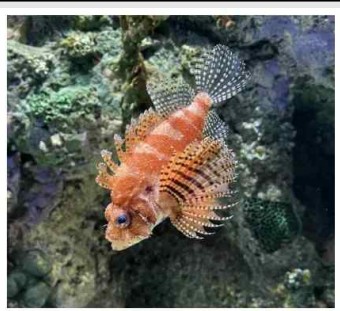

**Baseline**

The image features a close-up of a colorful fish with a long, thin tail and a large, pointy fin. The fish is swimming in a body of water, possibly an aquarium, with its head and body prominently visible. The fish appears to be a unique and eye-catching species, possibly a clownfish, given its distinctive appearance.

**Ours**

The image features a close-up of a colorful fish with a long, spiky tail, swimming in a tank. The fish has a vibrant orange and red color, with a distinctive orange and white striped pattern on its body. The fish's tail is long and spiky, adding to its unique appearance.

The tank is filled with various rocks, providing a natural and interesting environment for the fish. The rocks are scattered throughout the tank, with some located near the top, middle, and bottom of the image. The fish is positioned in the middle of the tank, surrounded by the rocks.

*Figure 8.* Comparison of captions generated by applying our method to LLaVA-1.5 and the baseline. Red text highlights incorrect references in the captions, while blue text indicates additional details provided by our method compared to the baseline.

## A.2. LLaVA-NeXT

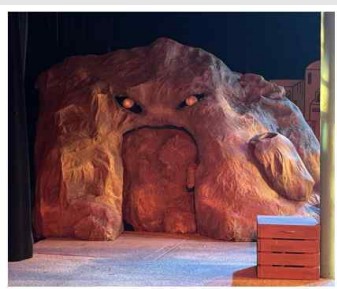

**Baseline**

The image captures a scene from a theatrical production, featuring a large, intricately designed rock formation as the centerpiece. The rock formation, which is the main focus of the image, is crafted from a combination of red and brown materials, giving it a rugged and natural appearance. It has a large opening in the center, inviting the viewer to explore what lies within. The rock formation is situated on a stage, which is bathed in a warm, orange light that enhances the dramatic effect of the scene. The stage is set against a black curtain, providing a stark contrast that further emphasizes the rock formation. In the foreground of the image, there are two wooden crates. Their presence suggests that they might be part of the set design or props for the production. The image does not contain any text or depict any actions. The relative positions of the objects are such that the rock formation is centrally located on the stage, with the wooden crates positioned in the foreground. The overall composition of the image suggests a scene from a theatrical performance, with the rock formation serving as a significant element in the narrative.

**Ours**

The image captures a scene from a theatrical production, featuring a large, rock-like prop that dominates the foreground. The prop, which appears to be made of a textured material resembling stone or wood, has a rough, uneven surface with a reddish-brown hue. It has two prominent, glowing eyes that are red and black, giving it a menacing appearance. The prop's mouth is open, revealing a dark interior, and there is a small, dark opening on the right side, possibly a door or entrance. In the background, there is a hint of a stage setting with a black curtain on the left side and a white structure that could be part of the set design. On the right side of the image, there is a red crate stacked on the floor, and a yellow light source is visible, illuminating the scene from the right side. The lighting casts shadows on the floor, enhancing the dramatic effect of the prop. The overall style of the image suggests a staged performance, possibly a play or a musical, with the prop serving as a significant element in the scene.

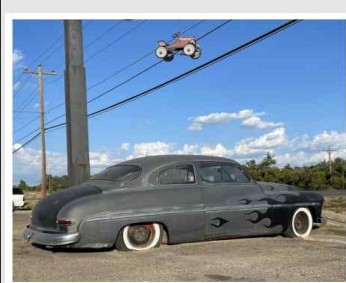

**Baseline**

The image captures a scene of a vintage car parked on a street, facing towards the right side of the frame. The car, painted in a shade of gray, is adorned with a flame design on its side, adding a touch of vintage charm. It's parked in front of a wooden telephone pole, which stands tall against the backdrop of a clear blue sky. The street on which the car is parked is lined with trees, their green leaves contrasting with the gray of the car. In the distance, power lines stretch across the sky, leading the eye towards the horizon. The image exudes a sense of nostalgia, reminiscent of a bygone era.

**Ours**

The image captures a scene of a vintage car parked on a gravel lot. The car, painted in a striking shade of black, is adorned with flames painted on its side, adding a touch of fiery spirit to its appearance. The car's design is reminiscent of the 1950s, with a rounded body and a curved roof, giving it a classic and timeless appeal. The car is parked on a gravel lot, which is scattered with small rocks and pebbles, adding texture to the scene. In the background, a telephone pole stands tall, its presence subtly reminding us of the era when such structures were more common. Above the car, a red toy tractor is suspended from power lines, adding an element of whimsy to the scene. The tractor, with its vibrant red color and playful design, contrasts with the seriousness of the vintage car below. The sky above is a clear blue, dotted with fluffy white clouds that add depth to the image. The power lines stretching across the sky create a sense of openness and freedom. Overall, this image paints a picture of a moment frozen in time, where vintage meets whimsy, and the past meets the present.

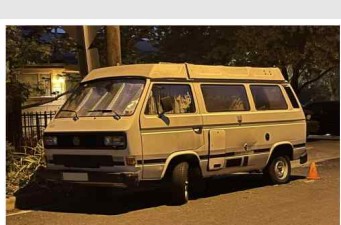

**Baseline**

The image depicts a beige Volkswagen van parked on a street at night. The van has a visible front grille with the Volkswagen logo, and the headlights are off. The vehicle is positioned on the side of the road, with the front facing towards the left side of the image. The van's side windows are tinted, and the rear window is clear. There is a visible dent on the driver's side door. The van is parked next to a curb, and there is a small orange traffic cone placed in front of it. The background shows a residential area with houses and trees, and the sky is dark, suggesting it is nighttime. There are no visible texts or distinctive markings on the van that provide additional information. The style of the image is a standard, unfiltered photograph with no artistic manipulation.

**Ours**

The image depicts a beige Volkswagen Type 2 van parked on a street at night. The van has a distinctive rounded front and a flat roof, which is typical of the Volkswagen Bus design. It features a black grille with the Volkswagen logo at the center, and the headlights are visible. The van has a white stripe running along the side, and the side windows are tinted. The vehicle is parked next to a curb, and there is a small orange traffic cone placed in front of the van. The background shows a residential area with houses and trees, and there is a street lamp visible in the distance. The lighting suggests it is nighttime, and the overall atmosphere is calm and quiet. There are no visible texts or distinctive markings on the van that provide additional information. The style of the image is a standard, unembellished photograph with no filters or artistic manipulations.

*Figure 9.* Comparison of captions generated by applying our method to LLaVA-NeXT and the baseline. Red text highlights incorrect references in the captions, while blue text indicates additional details provided by our method compared to the baseline.

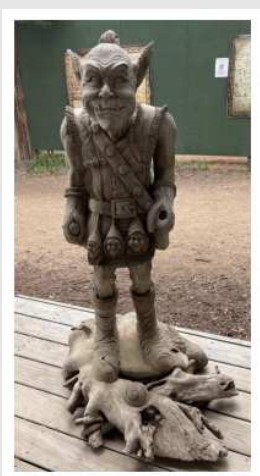

**Baseline**

The image shows a sculpture of a fantasy character, which appears to be a humanoid figure with a stylized, cartoonish design. The character has a large, exaggerated head with a prominent nose and a wide, cheerful smile. The figure is wearing a tunic with a belt and a pair of pants, and it has a large, round belly. The character's arms are raised, and it is holding what seems to be a small object in its right hand. The sculpture is standing on a base that resembles a log, with a large, curved piece of wood serving as the base. The background is out of focus, but it appears to be an outdoor setting with a green wall and a framed picture hanging on it. The ground is covered with dirt and scattered debris. There are no visible texts or distinguishing marks that provide additional context about the character or the setting. The style of the image is a straightforward, unembellished photograph with no filters or artistic effects applied.

**Ours**

The image shows a sculpture of a fantasy character, which appears to be a goblin or a similar creature, standing on a wooden platform. The sculpture is made of a light-colored material, possibly stone or concrete, and is detailed with textures and patterns that suggest a rugged, weathered appearance. The character has a large, pointed ear, a prominent nose, and a wide, grinning mouth. It is wearing a tunic with a belt and a pair of boots, and it is holding a small object in its right hand. The character is standing on a log that has been carved to resemble a stump, with a large knot visible on the surface.
The sculpture is placed outdoors, as indicated by the natural surroundings and the green wall in the background. There are two framed posters hanging on the wall, but the content of the posters is not visible in the image. The ground is covered with dirt and scattered debris, suggesting a natural or semi-natural setting. The lighting in the image is bright, and the photograph is taken from a slightly elevated angle, giving a clear view of the sculpture and its surroundings. There are no visible texts or brands in the image. The style of the image is a straightforward, unembellished photograph with no filters or artistic manipulations.

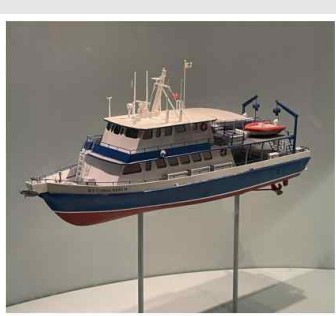

**Baseline**

The image shows a model of a blue and white boat with a red stripe on the hull, displayed on a stand. The boat has a prominent white superstructure with a cabin and a wheelhouse. There are several windows visible on the cabin and wheelhouse, and the boat is equipped with a lifeboat on the aft deck. The model is detailed with a flag flying from the mast and a lifebuoy hanging from the side. The background is a plain, light color, providing a neutral backdrop that highlights the model. There are no visible texts or brands on the model. The style of the image is a straightforward product photograph with a focus on the model boat, intended to showcase its design and features.

**Ours**

The image shows a model of a blue and white boat with a red stripe on the hull, displayed on a stand. The boat has a prominent white superstructure with a cabin and a wheelhouse, and a blue and white hull with a red stripe running along the side. On the bow, there is a red lifeboat with a white interior. The model is mounted on a white stand with two vertical poles supporting it. The background is a plain, light gray surface, and there is a small, circular object with a flag on top of the model, suggesting it might be a decorative element or a part of the model's display. The text "RV Coral Reef II" is visible on the side of the model, indicating the name of the boat. The style of the image is a straightforward, clear photograph with no additional context or embellishments.

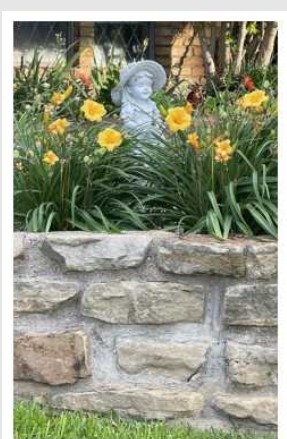

**Baseline**

The image shows a garden scene with a focus on a statue of a person wearing a straw hat, positioned in the center. The statue is partially obscured by lush greenery, including a variety of flowers and plants. The statue appears to be made of a light-colored material, possibly stone or concrete, and is set against a backdrop of a brick wall. The wall is constructed of rectangular bricks and has a rough texture. The garden is bordered by a neatly trimmed lawn on the left side, and there are trees and shrubs in the background, adding to the natural setting. The overall style of the image is a realistic photograph with a focus on the garden and the statue. There are no visible texts or distinctive brands in the image.

**Ours**

The image shows a garden scene with a focus on a statue of a young girl with a straw hat, positioned in the center. The statue is made of a light-colored material, possibly stone or concrete, and is partially obscured by lush green foliage. The girl is depicted with a serene expression, looking towards the viewer. Surrounding the statue are various yellow flowers, including daisies and possibly other types of flowers, which add a vibrant splash of color to the scene. The flowers are in full bloom, suggesting a warm, possibly spring or summer season.

The garden bed is bordered by a low stone wall, which appears to be made of stacked stones, giving it a rustic and natural look. The wall is partially covered by the green foliage, blending the garden's elements together. In the background, there is a brick wall, which provides a contrasting texture to the softness of the garden bed. The overall impression is one of a tranquil and well-maintained garden space. There are no visible texts or distinctive branding in the image. The style of the image is a realistic photograph with a focus on the garden and its decorative elements.

*Figure 10.* Comparison of captions generated by applying our method to LLaVA-NeXT and the baseline. Red text highlights incorrect references in the captions, while blue text indicates additional details provided by our method compared to the baseline.

# B. Details for Analyses

## B.1. Why More Attention Doesn't Always Mean Better Descriptions

**Enhanced Attention: A Path to Less Diversity?**    To analyze the diversity of visual attention during caption generation, we used the baseline model (LLaVA-1.5 7B) to generate captions for 3,000 image samples from the DOCCI dataset. During the caption generation process, we measured the attention weights assigned to visual tokens when generating output tokens. Specifically, we extracted visual attention weights from layer 20 and averaged them across attention heads.

For each image sample, we analyzed the visual attention corresponding to the first 100 output tokens. To quantify the variation in visual attention throughout the generation process, we computed the distance between the visual attention distributions of consecutive output tokens within each sample. The distance was calculated using the Wasserstein distance, where each visual attention distribution was first normalized so that the sum of attention weights across all tokens equaled 1.

The results of this analysis are presented in Figure 3, which compares the baseline model with a naive attention enhancement method that proportionally increases visual attention weights. The plot illustrates that the naive enhancement method reduces the variation in visual attention patterns throughout the caption generation process. This finding aligns with our observations that the naive method decreases the diversity of objects mentioned in the generated captions.

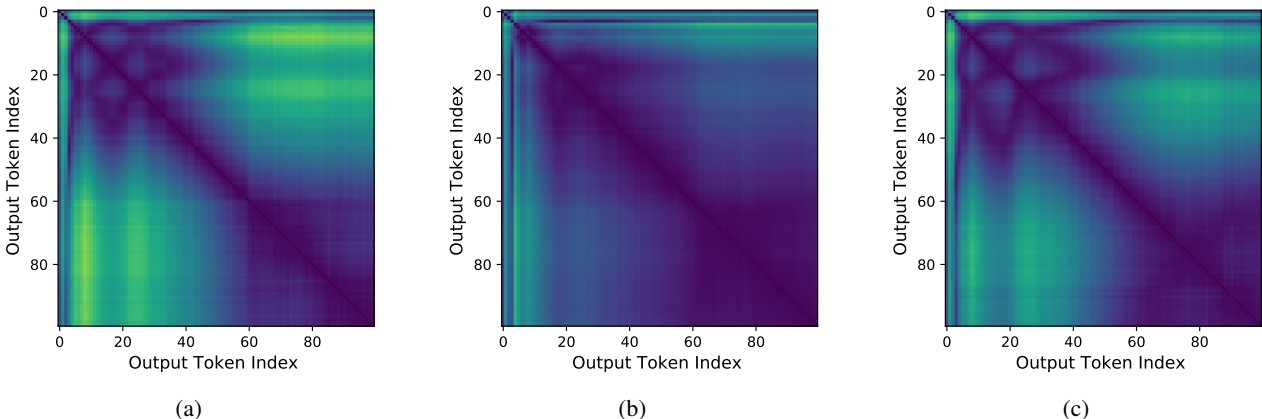

(a)                                         (b)                                         (c)

*Figure 11.* Visual attention diversity comparison between (a) the baseline model, (b) the naive attention enhancement approach, and (c) our method. The naive approach reduces visual attention diversity, indicating ineffective adaptation to important visual tokens. Instead, our proposed method does not severely degrade visual attention diversity

In contrast, the results shown in Figure 11 demonstrate that our proposed method does not severely degrade visual attention diversity. Instead, our approach dynamically reinforces attention to relevant regions of an image based on the evolving context during caption generation. This confirms that our method effectively maintains a balance between enhancing visual attention and preserving diversity, leading to more contextually appropriate and varied caption outputs.

In addition to its impact on visual attention, our method also preserves diversity in the content of generated captions. Appendix D presents an analysis of caption diversity, introducing metrics that assess the variation in generated sentences. Using these metrics, we evaluate caption diversity across different methods, including those discussed earlier in this appendix.

**Longer Contexts Amplify Noisy Attention**    To analyze how longer contexts influence attention noise, we conducted an experiment using the baseline model (LLaVA-1.5 7B). During the caption generation process for a single image, we measured the attention weights assigned to visual tokens when generating output tokens. Specifically, visual attention weights were extracted from layer 20 and averaged across attention heads.

Next, we normalized the visual attention weights such that the sum of all attention weights for visual tokens equaled 1. Finally, as shown in Figure 4, we plotted the normalized attention distribution along with the corresponding image to visualize how attention shifts across different tokens in the presence of longer contexts.

Additionally, To address the concern that raw attention scores—especially in deeper layers—may not directly correspond to the original image patches, we conducted additional analyses using saliency maps computed with respect to the attention scores. Specifically, we implemented a gradient-weighted attention method inspired by (Zhang et al., 2025), where attention scores are weighted by their gradients with respect to the model's output.

As show in Figure 12, we compare the original attention-based results (Figure 4) with those obtained using the saliency maps. The saliency maps reveal similar overall trends: as the context length increases during caption generation, the resulting maps become progressively noisier. This observation supports our original interpretation. Furthermore, recent work (Zhang et al., 2025) has demonstrated that the visual attention pattern of MLLMs do indeed align with semantically relevant image regions, particularly in tasks like visual question answering, further validating the use of this method in our analysis.

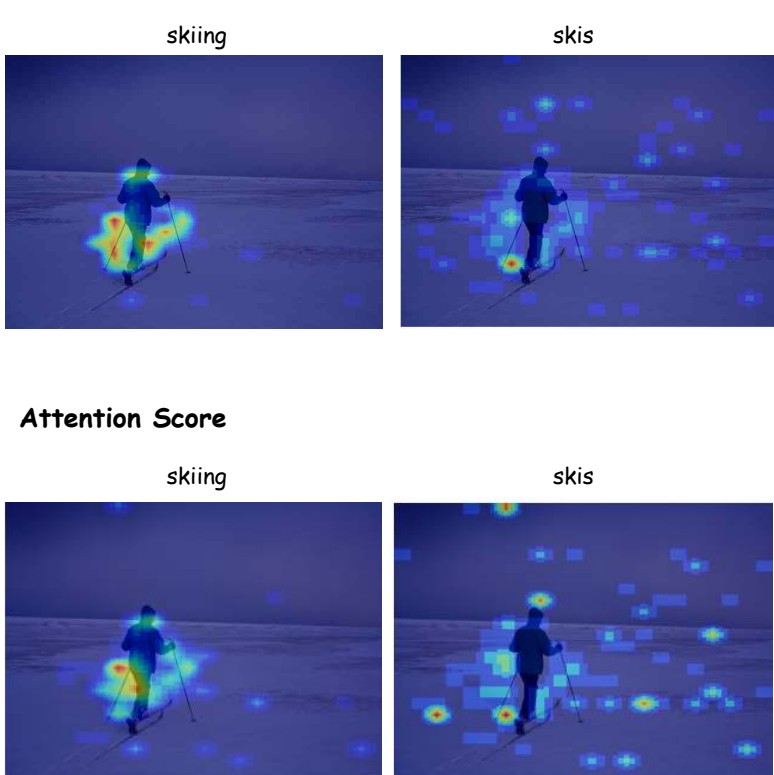

Figure 12. Comparison between visualizations of attention scores and gradient-weighted attention scores for different context lengths during caption generation. Both methods exhibit similar trends, with increasing noise as the context length grows.

**Longer Context, Less Visual Focus**   To analyze the effect of longer contexts on visual focus, we conducted an experiment using the baseline model (LLaVA-1.5 7B). Caption generation was performed on 3,000 image samples from the DOCCI dataset, and we measured the attention weights assigned to visual tokens during the generation of output tokens.

Specifically, visual attention weights were extracted from layer 20 and averaged across attention heads. For each output token, we computed the total attention weights allocated to visual tokens and the total attention weights allocated to text tokens (including instruction tokens and generated tokens). These values were averaged across all samples and plotted against different context lengths to examine how visual focus changes as context length increases. The results of this analysis are presented in Figure 5, which illustrates the diminishing visual focus as context length grows.

Figure 5 does not normalize for the total number of tokens, which may obscure the disproportionate decline in attention to visual information as the caption length increases—especially since the number of text tokens naturally grows with longer

captions. To address this, we additionally analyze the average attention per token by dividing the total attention to image and text tokens by their respective token counts. As shown in Figure 13, this normalized analysis reveals that attention to image tokens decreases disproportionately faster than to text tokens as the context length increases.

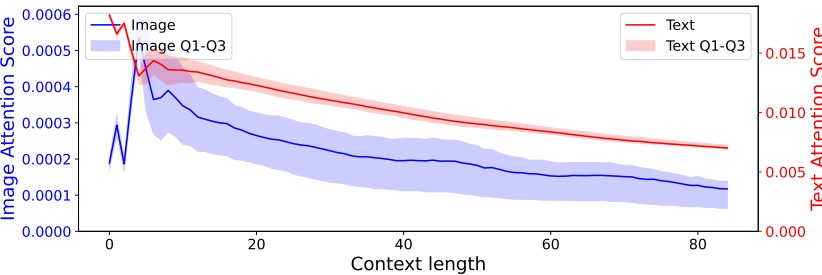

*Figure 13.* Average attention weight trends for text and image tokens as a function of context length during caption generation, computed by dividing attention by the number of respective tokens. This normalized view reveals that attention to image tokens decreases more sharply than to text tokens as context length increases.

## B.2. Analyses in Section 5.3

**Visual Attention Analysis**    To analyze visual attention during caption generation, we conducted an experiment using the baseline model (LLaVA-1.5 7B) on 3,000 image samples from the DOCCI dataset. During the caption generation process, we measured the attention weights assigned to visual tokens when generating output tokens.

Specifically, visual attention weights were extracted from layer 20 and averaged across attention heads. We computed the total attention weights allocated to visual tokens for each output token and compared the results between the baseline model and the SPARC-enhanced model. These values were plotted to illustrate the differences in visual attention between the two approaches, as shown in Figure 7(a).

**Impact on attention sinks**    To evaluate whether each method strengthens visual attention to contextually relevant regions or amplifies unrelated areas, we analyzed how different approaches affect attention sinks—tokens that receive large attention values despite being unrelated to the actual context.

Specifically, we measured the ratio of attention scales by computing the average attention weight of sink tokens divided by the average attention weight of non-sink tokens. This ratio was used to assess the extent to which each method influences attention sinks, as illustrated in Figure 7(b).

For this experiment, we used the baseline model (LLaVA-1.5 7B) to generate captions for 3,000 image samples from the DOCCI dataset. During caption generation, we measured the attention weights assigned to visual tokens at layer 20 and averaged them across attention heads.

To identify visual attention sinks, we followed the approach described in previous work (Kang et al., 2025). Specifically, attention sinks were determined based on high-dimensional hidden states (self-attention layer inputs), where certain dimensions exhibited significantly large values. Tokens with high values in these specific dimensions were classified as sink tokens.

During generation, for each output token, we computed the ratio of average attention weights between sink tokens and non-sink tokens. These values were then averaged across all image samples and plotted for comparison across the baseline model, a naive attention enhancement approach, and SPARC.

Results indicate that the naive approach significantly increases the proportion of attention allocated to sink tokens, while SPARC maintains the sink attention ratio at similar levels to the baseline. Additionally, prior results (Figure 7(a)) demonstrated that SPARC counteracts the decrease in attention values caused by longer context lengths. Taken together, these findings suggest that SPARC selectively reinforces attention to important visual tokens as context length increases, rather than indiscriminately amplifying all attention values.

# C. Ablation Study on the Effectiveness of Hyperparameter and Setting Varitions

**Ablation Study on Setting Choices**    To demonstrate the effectiveness of the proposed components in our method, we conducted a series of ablation experiments. Specifically, we evaluated the impact of our token selection strategy and the progressive attention calibration mechanism, as presented in Table 7. For these experiments, we used LLaVA-1.5 as the baseline model and measured performance using the CLAIR metric on the IIW-400 dataset.

First, we examined the effect of omitting the token selection process and applying the progressive attention calibration to all image tokens. To ensure a fair comparison, we set the attention scaling factor $\alpha$ to 1.007, following the same scaling strategy as when token selection is employed, as shown in Figure 7(a). While this approach led to performance improvements compared to the baseline, demonstrating the effectiveness of the progressive attention calibration mechanism, the results fell short of the performance achieved by our complete method. This indicates that while progressive attention calibration contributes to performance gains, the token selection strategy further enhances the model's ability to focus on the most informative tokens, leading to superior overall performance.

Next, we analyzed the effect of modifying the token selection approach by comparing the tokens to those from only the immediately preceding step, rather than leveraging the exponential moving average (EMA) across previous steps. This simplified selection mechanism yielded better performance than the baseline but did not match the effectiveness of our proposed method.

These ablation results underscore the contributions of both the token selection strategy and the progressive attention calibration in enhancing model performance. Our full method effectively mitigates attention dilution and ensures the consistent integration of salient visual information throughout the caption generation process.

*Table 7.* CLAIR scores under different setting choices.

| SETTING | CLAIR |
|---|---|
| BASELINE | 56.35 |
| OURS | 61.49 |
| w/o SELECTION | 59.10 |
| w/o EMA | 60.28 |

**Ablation Study on Hyperparameter Impact**    To further analyze the effectiveness of the hyperparameters in our method, we conducted a series of experiments by systematically varying key parameters and evaluating their impact on performance. Specifically, we assessed how changes to individual hyperparameters influenced the model's performance, using LLaVA-1.5 as the baseline model and measuring the CLAIR score on the IIW-400 dataset. The parameters examined in this study were those introduced in Section 5.1.

First, we evaluated the impact of the layer $l$ at which token selection is applied. Regardless of the layer at which token selection was performed and subsequent attention recalibration was applied, we observed an improvement in CLAIR scores compared to the baseline. Notably, selecting tokens in mid-to-late layers resulted in the most significant performance gains, with the highest CLAIR score observed at layer 20 (Table 8). This finding aligns with prior research, which has shown that the visual token attention patterns in MLLMs tend to align more closely with semantically meaningful features in mid-to-late transformer layers (Jiang et al., 2024).

Next, we investigated the impact of the token selection threshold $\tau$, which determines the relative activation score used for token selection (Equation (5)). This score quantifies how much the attention on a given visual token jumps compared to the previous output token during caption generation. The results, presented in Table 8, reveal that excessively low threshold values degrade captioning performance. However, for appropriately chosen values, reinforcing attention on the selected tokens consistently led to improvements over the baseline. The best performance was observed at $\tau = 1.5$, where the model achieved the highest CLAIR score.

Then, we evaluated the impact of the EMA (exponential moving average) smoothing factor $\beta$, which controls the degree to which historical attention patterns influence token selection. The results, shown in Table 8, indicate that when $\beta = 0$, the model relies solely on the difference between the current step's attention and that of the immediately preceding step. In contrast, applying EMA smoothing enables a more stable token selection process. We found that using a small value of $\beta = 0.1$ was particularly effective, as it allowed token selection to be influenced primarily by the most recent step while still

incorporating a degree of historical information.

Finally, we conducted additional ablation experiments on the scaling parameter $\alpha$. We investigated how increasing $\alpha$ affects CLAIR scores and shown in Table 8, found that performance improved up to a certain threshold, with the highest CLAIR score observed at $\alpha = 1.1$. However, increasing $\alpha$ beyond this point degraded the model's captioning performance.

*Table 8.* CLAIR scores according to different hyperparameters for LLaVA.

| PARAMETER | | | | | | |
|---|---|---|---|---|---|---|
| **LAYER** | 5 | 10 | 15 | **20** | 25 | 30 |
| CLAIR | 57.34 | 58.18 | 58.30 | **61.49** | 60.85 | 58.29 |
| $\tau$ | 0.5 | 1.0 | **1.5** | 2.0 | 2.5 | |
| CLAIR | 52.03 | 58.56 | **61.49** | 60.24 | 59.50 | |
| $\beta$ | 0.3 | 0.2 | 0.15 | **0.1** | 0.05 | 0 |
| CLAIR | 60.00 | 60.10 | 61.20 | **61.49** | 60.41 | 60.28 |
| $\alpha$ | 1.05 | 1.075 | **1.1** | 1.125 | 1.15 | |
| CLAIR | 58.60 | 60.06 | **61.49** | 60.90 | 57.93 | |

Furthermore, we extended the ablation studies to additional models and datasets beyond LLaVA-1.5. Specifically, we applied the same parameter variation framework to LLaVA-NeXT and Qwen2-VL, using the DOCCI dataset for evaluation. We randomly sampled 500 images and generated captions for each model, and subsequently evaluating the outputs using the CLAIR score.

Following the same protocol as in the LLaVA-1.5 ablation, we systematically varied four key parameters—token selection layer $l$, token selection threshold $\tau$, EMA smoothing factor $\beta$, and scaling parameter $\alpha$—and measured their respective impact on performance. Table 9 presents the results for LLaVA-NeXT, while Table 10 shows the results for Qwen2-VL. As reported in Table 4, the configuration used for LLaVA-NeXT was $l = 20$, $\tau = 4$, $\alpha = 1.1$, $\beta = 0.1$ with a CLAIR score of 66.99, and for Qwen2-VL was $l = 18$, $\tau = 3$, $\alpha = 1.1$, $\beta = 0.1$ with a CLAIR score of 80.64.

*Table 9.* CLAIR scores according to different hyperparameters for LLaVA-NeXT.

| PARAMETER | | | | | |
|---|---|---|---|---|---|
| **LAYER** | 10 | 15 | **20** | 25 | 30 |
| CLAIR SCORE | 63.97 | 65.25 | **66.99** | 64.61 | 63.58 |
| $\tau$ | 2.5 | **3.0** | 3.5 | 4.0 | |
| CLAIR SCORE | 68.10 | **68.60** | 67.81 | 66.99 | |
| $\beta$ | 0.2 | 0.15 | **0.1** | 0.05 | 0.0 |
| CLAIR SCORE | 65.93 | 66.34 | **66.99** | 66.57 | 67.80 |
| $\alpha$ | 1.05 | 1.075 | **1.1** | 1.125 | |
| CLAIR SCORE | 64.20 | 65.10 | **66.99** | 67.26 | |

Across all models (LLaVA-1.5, LLaVA-NeXT, Qwen2-VL), we observe similar trends regarding the optimal range for the layer, $\alpha$, $\beta$ and $\tau$ parameters—typically favoring mid-to-late transformer layers and slightly scaled values. Importantly, our method is training-free and incurs minimal additional computational overhead, making it both practical and efficient for lightweight hyperparameter tuning in real-world scenarios. While some degree of tuning is required to achieve optimal performance for each model, we highlight that effective settings can be identified with relative ease due to the simplicity and efficiency of our approach.

To further analyze the effect of $\alpha$, we conducted additional ablation experiments using the CHAIR benchmark, measuring Precision, Recall, and F1 score at the object level based on the caption content as $\alpha$ varied. The results are presented in Figure 14, where we evaluated both LLaVA-1.5 and LLaVA-NeXT models. Figure 14(a) shows the results for LLaVA-1.5, where performance improved across Precision, Recall, and F1 scores as $\alpha$ increased up to 1.1. At $\alpha = 1.12$, Recall slightly

*Table 10.* CLAIR scores according to different hyperparameters for Qwen2-VL

| **PARAMETER** | | | | | |
|---|---|---|---|---|---|
| **LAYER** | 10 | **18** | 20 | 28 | |
| CLAIR SCORE | 79.62 | **80.64** | 79.36 | 79.54 | |
| $\tau$ | 2.0 | 2.5 | **3.0** | 3.5 | |
| CLAIR SCORE | 77.99 | 78.98 | **80.64** | 79.77 | |
| $\beta$ | 0.2 | 0.15 | **0.1** | 0.05 | 0.0 |
| CLAIR SCORE | 80.36 | 79.52 | **80.64** | 79.76 | 79.61 |
| $\alpha$ | 1.05 | 1.075 | **1.1** | 1.125 | |
| CLAIR SCORE | 80.31 | 80.14 | **80.64** | 78.85 | |

decreased compared to the baseline, but Precision improved significantly. Interestingly, at this setting, the Precision level was similar to that of the naive attention enhancement approach (Liu et al., 2025) in Table 2, while maintaining a higher Recall, suggesting that our method provides a better balance in the Precision-Recall trade-off. Figure 14(b) presents the results for LLaVA-NeXT, where Precision remained close to the baseline, while Recall increased, indicating that our approach enhances caption quality by improving recall without sacrificing precision.

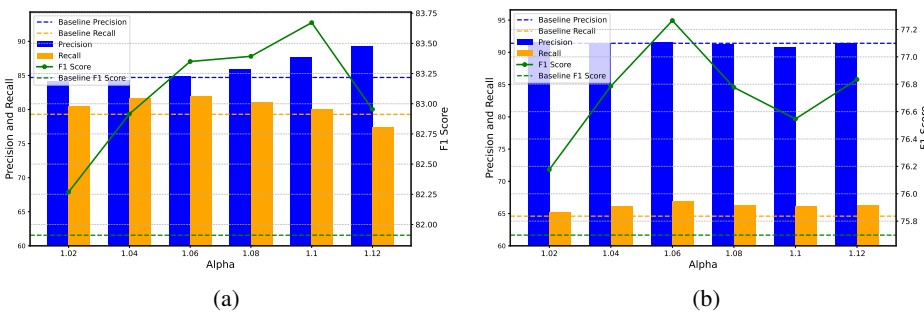

(a)                       (b)

*Figure 14.* Object-level Precision, Recall, and F1 scores for LLaVA-1.5 (a) and LLaVA-NeXT (b) with scaling parameter $\alpha$

## D. Detailed Caption Similarity Analysis

In the previous sections, we observed that naively enhancing the attention on visual tokens leads to a reduction in the number of objects mentioned in the generated captions and decreases the diversity of attention patterns across visual tokens during the captioning process. To further investigate the relationship between the diversity of attention patterns and the variety of objects mentioned in the generated captions, we propose a metric that quantifies the similarity between sentences within the generated captions. Using this metric, we evaluate the caption diversity of different methods and analyze their effects. Our experimental results demonstrate that naive attention amplification strategies increase the similarity between sentences within captions, thereby reducing overall caption diversity. Furthermore, we show that our proposed token selection method directly enhances the diversity of objects mentioned in captions, supporting the significance of our approach.

**Caption Similarity Anlaysis** To quantify this effect, we introduce the **Caption Similarity Score** $C_{sim}$, which measures the similarity between pairs of sentences within a generated caption. It is defined as:

$$C_{sim} : \frac{1}{|P|} \sum_{(s_i, s_j) \in P} S_{sim}(s_i, s_j) \tag{10}$$

where $P$ denotes the set of all possible sentence pairs within a given caption. Each pair $(s_i, s_j) \in P$ consists of two sentences $s_i, s_j$, and $S_{sim}(s_i, s_j)$ represents their similarity score. Specifically, $S_{sim}$ is calculated using a large language model (LLM), which evaluates the degree of overlap in visual content described by the paired sentences. A higher $C_{sim}$ score indicates greater similarity between sentences, suggesting lower diversity in the generated caption.

For our experiments, we generated captions for images sampled from the IIW-400 dataset and computed the caption similarity scores using LLaMA-3.2-11B-Vision (Dubey et al., 2024). The prompt used for measuring similarity is as follows:

```
You are trying to tell if two sentences are describing the same visual contents.
Sentence1: {sentence1} Sentence2: {sentence2} On a precise scale from 0 to 100,
how likely is it that the two sentences are describing the same visual contents?
```

This prompt is similar to the one used in CLAIR to compare the generated caption and the reference caption.

To validate the impact of our approach, we measured $C_{sim}$ for captions generated by the baseline model, the naive attention-enhanced method, and our proposed method. As illustrated in Figure 15(a), our results indicate that naive attention amplification strategies, while intended to enhance focus across the entire image, inadvertently lead to higher caption similarity scores. This suggests that the generated captions tend to repeat descriptions of the same visual elements, thereby reducing the diversity of objects and features mentioned, limiting the informativeness and variety of the captions. In contrast, our method increases the diversity of captions by reducing $C_{sim}$, demonstrating its effectiveness in improving caption variability.

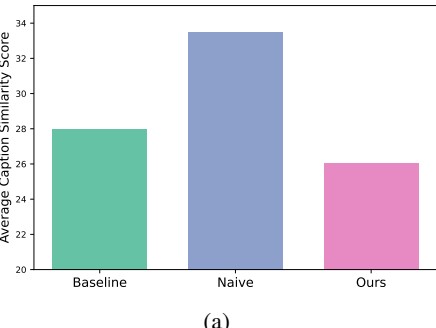 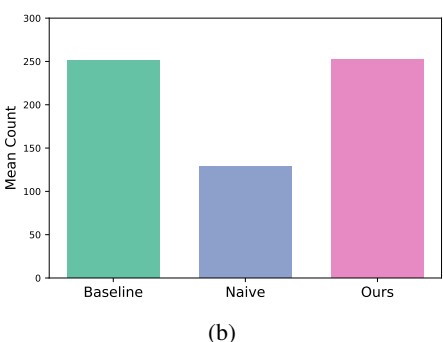

(a)  (b)

*Figure 15.* (a) Comparison of caption similarity scores and number of selected tokens. (a) Caption similarity score across different methods. (b) Number of dynamically selected tokens during caption generation. Methods that identify more tokens as important tend to yield lower similarity scores, indicating that focusing on a broader range of tokens increases diversity in the generated descriptions.

**Impact of Token Selection on Caption Similarity**  We further investigate whether the visual tokens selected by our method dynamically align with the most relevant visual content for the generated text. Specifically, we examine how frequently each image token is selected during the captioning process to assess its contextual significance. For each caption in the dataset, visual tokens are chosen at each generation step using the **Relative Activation Score** ( Section 4.2). To quantify the selection dynamics, we used a cumulative **Selection Count** in Equation (7), which represents the total number of times each token has been selected throughout the caption generation process.

UWith this metric, we assess the extent of dynamic token selection by categorizing tokens according to their total selection count throughout the caption-generation process. We classify tokens into two groups: frequently selected tokens (high selection count) and infrequently selected tokens (low selection count). By comparing the number of tokens in the high-selection group across different methods, we evaluate how different approaches influence the degree of dynamic visual token activation during captioning.

We analyze the selection dynamics across 3,000 image samples from the DOCCI dataset by measuring the number of tokens in the high-selection group during the captioning process and computing the average. The results are presented in Figure 15(b).

Interestingly, methods that dynamically select a larger number of tokens throughout the process tend to produce captions with lower similarity scores. This suggests that a higher degree of dynamic visual token selection contributes to greater variability in generated captions. This aligns with the intuition that focusing on a broader range of tokens increases diversity in the generated descriptions. The naive approach may restrict the model to a narrow set of tokens, thereby limiting caption diversity. In contrast, our proposed selection mechanism remains flexible, emphasizing contextually important tokens without over-constraining the attention space. This flexibility enables richer and more coherent visual grounding in the final

captions, overcoming the limitations of earlier approaches.

In summary, our findings demonstrate that naive attention-enhancement strategies inadvertently reduce caption diversity by increasing similarity between sentences. In contrast, our proposed token selection method enhances diversity by dynamically selecting a broader set of visually relevant tokens. These results highlight the importance of effective token selection in generating informative and diverse captions.

## E. Evaluation on Other Hallucination Benchmark

While our primary focus is on enhancing detailed image captioning, we also performed additional experiments on the POPE hallucination benchmark (Li et al., 2023b). Inspired by the evaluation setup proposed in (Mitra et al., 2024), where caption-based reasoning improves MLLM performance on general multimodal tasks, we adopt a similar strategy. Specifically, we prompted the model to first generate a caption for the image and then use it as part of the input to answer the question.

We evaluated Qwen2-VL (7B) using three approaches: the baseline model, our proposed method, and naive attention scaling (PAI) (Liu et al., 2025). For the naive method, we identified the optimal hyperparameter ($\alpha = 0.2$) before evaluation.

The table below summarizes the accuracy on the POPE benchmark, considering only those responses that adhered to the instruction by including both a caption and an answer.

*Table 11.* Accuracy on the POPE hallucination benchmark.

| Method | Accuracy (%) |
| --- | --- |
| Baseline | 82.01 |
| Naive Attn. Scaling | 81.45 |
| Ours | **83.13** |

Our method shows an improvement over the baseline, while the naive attention scaling slightly reduces accuracy. Additionally, we evaluated the instruction-following behavior, measured as the proportion of responses in which the model correctly generates an output that includes both a caption and an answer.

*Table 12.* Instruction-following behavior on the POPE benchmark.

| Method | Instruction Following (%) |
| --- | --- |
| Naive Attn. Scaling | 76.84 |
| Ours | 92.72 |

These results indicate that naive attention scaling may reduce the model's sensitivity to the input prompt, whereas our method retains alignment with instruction while improving grounding accuracy.

## F. CHAIR Metric Results and Precision-Recall Analysis

To provide a more detailed analysis of hallucination and precision-recall tradeoff, we present the CHAIR metric results along with recall, and F1 score. The CHAIR metric consists of two components:

**Instance-level hallucination rate (CHAIR$_i$)**

$$\text{CHAIR}_i = \frac{|\{\text{hallucinated objects}\}|}{|\{\text{all objects mentioned}\}|} \tag{11}$$

This metric measures the proportion of hallucinated objects among all mentioned objects in generated captions. Since precision in our primary evaluation is defined as $1 - \text{CHAIR}_i$, a lower $\text{CHAIR}_i$ value directly translates to a higher precision, indicating fewer hallucinated objects in the generated captions.

**Caption-level hallucination rate (CHAIR$_s$)**

$$\text{CHAIR}_s = \frac{|\{\text{captions with hallucinated object}\}|}{|\{\text{all sentences}\}|} \tag{12}$$

This metric captures the proportion of captions containing at least one hallucinated object. A lower $\text{CHAIR}_s$ value suggests that hallucinations are less frequent across generated captions at the caption level.

Table Table 13 summarizes the evaluation scores, where each value represents the average over five repeated experiments on randomly sampled 500 instances.

Our method adopts a progressive reinforcement mechanism that enhances attention to visual tokens as the caption generation progresses. This strategy helps mitigate hallucinations in the latter part of the caption, leading to a noticeable improvement over baseline methods in later-token precision. However, since this approach primarily addresses hallucinations that occur later in the caption, it is inherently less effective in handling hallucinations that appear early in the sequence. As a result, our method may still exhibit limitations when evaluated under metrics such as $\text{CHAIR}_s$, which assess hallucination at the overall caption level.

Nevertheless, despite this inherent challenge, our approach demonstrates superior performance in F1 score by achieving a better balance between reducing hallucination (increasing precision) and improving recall. These results indicate that our method successfully enhances the model's ability to generate captions that are both accurate and comprehensive, establishing a new standard for high-quality captioning.

*Table 13.* Comparison of CHAIR metric results on the MS COCO validation dataset for existing methods and our proposed approach. Our method achieves a higher F1 score by effectively mitigating hallucination while maintaining recall, demonstrating a better balance compared to other approaches.

| METHODS | CHAIR$_s$ | CHAIR$_i$ | RECALL | F1 |
|---|---|---|---|---|
| BASELINE | 53.96 | 15.30 | 79.46 | 81.99 |
| OPERA | 55.68 | 15.46 | 78.82 | 81.58 |
| VCD | 57.92 | 16.78 | 77.50 | 80.26 |
| VOLCANO | 46.16 | 12.46 | 77.82 | 82.39 |
| PAI | **34.92** | **9.36** | 72.44 | 80.52 |
| OURS | 51.52 | 12.28 | **79.98** | **83.67** |

