# OpenReview forum: "Visual Attention Never Fades: Selective Progressive Attention ReCalibration for Detailed Image Captioning in Multimodal Large Language Models"
_ICML.cc/2025/Conference — ICML 2025 poster_

### Official Review · Reviewer_4gUZ · 2025-03-13

**Overall Recommendation:** 3

**Summary:**

This paper focuses on improving detailed image captioning quality in VLMs. The authors argue that existing models struggle to maintain strong visual attention when generating longer captions, causing increased noise and reduced recall. To fix this, they propose a method that selectively strengthens visual attention by tracking significant changes in attention values over time. The method also reinforces consistently important visual tokens during decoding.

**Claims And Evidence:**

- The authors claim their method improves visual attention quality, but the paper only shows results on captioning benchmarks. There’s no direct evidence proving attention actually got better. The attention maps provided still look pretty noisy, both before and after applying their method.
- They mention “minimal computational overhead” but don’t back it up with runtime numbers or memory comparisons, so this efficiency claim isn’t convincing.

**Essential References Not Discussed:**

N/A

**Experimental Designs Or Analyses:**

- Benchmarks used (CLAIR, CHAIR, human evaluation) are appropriate and sufficient
- Hyperparameter settings (α, β, τ) are arbitrarily chosen without sensitivity analysis, weakening the reliability of experimental results.
- Human evaluation uses only 100 examples, limiting generalizability and robustness.
- Computational overhead is claimed minimal, but no quantitative runtime or memory analysis is presented to support this.

**Methods And Evaluation Criteria:**

The benchmarks (CLAIR, CHAIR) and datasets (IIW-400, DOCCI, MS-COCO) used to evaluate caption quality are standard and appropriate. However, the evaluation is a bit incomplete—the authors rely solely on extrinsic metrics and provide no convincing intrinsic evaluation (e.g., clearer attention visualizations or quantitative metrics) to directly validate their claim that attention quality improved.

**Other Comments Or Suggestions:**

N/A

**Other Strengths And Weaknesses:**

N/A

**Questions For Authors:**

N/A

**Relation To Broader Scientific Literature:**

This paper studies detailed caption generation in VLMs, connecting closely to previous work on improving attention mechanisms in vision-based models, particularly techniques that aim to enhance visual attention quality. It also aligns with research addressing hallucination in natural language generation, particularly within text summarization, where maintaining factual consistency and reducing irrelevant or incorrect content is a common challenge.

**Theoretical Claims:**

N/A

---

> ### Author Rebuttal · Authors · 2025-04-01
>
> We sincerely thank the reviewer for the detailed and helpful review. We apologize for any confusion caused by the incomplete results in the original submission. Your comments have been very valuable, and we provide our detailed responses below. We would also appreciate any further suggestions or feedback.
>
> **1. Quantitative Evaluation for Attention Quality**
>
> Thank you for raising this important point. We agree that intrinsic, quantitative evidence is necessary to directly support our claims regarding attention quality.
>
> To this end, we analyzed **visual attention** by measuring how much attention is assigned to **semantically relevant image regions** during the decoding process, using 5,000 randomly sampled images from the MSCOCO 2014 validation set. Specifically, for each generated token corresponding to a ground truth object in the caption, we calculated the **total attention score allocated to image tokens** within the region of that object.
>
> To identify these regions, we used an **open-vocabulary segmentation model (Grounded SAM2 [1])** to generate binary masks for all ground truth objects. During caption generation, for each object word token, we measured the **proportion of visual attention focused on the corresponding object region** out of the total attention score across all image tokens.
>
> The table below summarizes the results across three methods:
>
> | Method | Attention on Relevant Image Regions (%) |
> | --- | --- |
> | Baseline | 17.85 |
> | Ours | 19.17 |
> | Naive Attention Scaling | 15.50 |
>
> These results indicate that our method achieves better alignment between generated text and relevant visual regions compared to the baseline and naive attention scaling. This suggests that our model's visual attention is **less noisy** and more focused on semantically meaningful parts of the image during captioning.
>
> [1] Ren, T., Shen, S. Grounded SAM 2: Ground and Track Anything in Videos. IDEA-Research, GitHub, 2024.
>
> ---
> **2. Computational Efficiency Comparison**
>
> Thank you for your comment. We agree that empirical evidence is important to support our efficiency claims. A detailed comparison of runtime and memory overhead—including per-token generation latency and storage requirements—is provided in our response to **Reviewer JgJt**. To avoid redundancy, we kindly refer you to that section for a full breakdown. In brief, our method introduces only minimal overhead compared to the baseline, while remaining significantly more efficient than prior approaches.
>
> ---
> **3. More Ablations for Parameters**
>
> Thank you for raising this important concern regarding hyperparameter sensitivity. As mentioned in the main paper, we initially conducted ablation studies on hyperparameters using the **LLaVA-1.5 (7B)** model on the **IIW-400** dataset. The results are presented in Tables 6–9 of the supplementary material.
>
> To further strengthen our analysis and address the reviewer’s concern, we extended these ablations to additional models and datasets. Specifically, we conducted experiments using **LLaVA-NeXT (7B)** and **Qwen2-VL (7B)** on the **DOCCI** dataset. We randomly sampled 500 images and evaluated the quality of the generated captions using the **CLAIR score**. The results are summarized below.
>
> **LLaVA-NeXT (7B)**
>
> Baseline CLAIR score: **62.49**
>
> Ours (Layer 20, $\tau$=4, $\alpha$=1.1, $\beta$=0.1): **66.99**
>
> | Layer | 10 | 15 | 20 | 25 | 30 |
> | --- | --- | --- | --- | --- | --- |
> | Score | 63.97 | 65.25 | 66.99 | 64.61 | 63.58 |
>
> | $\tau$ | 2.5 | 3.0 | 3.5 | 4.0 |
> | --- | --- | --- | --- | --- |
> | Score | 68.10 | 68.60 | 67.81 | 66.99 |
>
> | $\alpha$ | 1.05 | 1.075 | 1.1 | 1.125 |
> | --- | --- | --- | --- | --- |
> | Score | 64.20 | 65.10 | 66.99 | 67.26 |
>
> | $\beta$ | 0.2 | 0.15 | 0.1 | 0.05 | 0.0 |
> | --- | --- | --- | --- | --- | --- |
> | Score | 65.93 | 66.34 | 66.99 | 66.57 | 67.80 |
>
> **Qwen2-VL (7B)**
>
> Baseline CLAIR score: **79.22**
>
> Ours (Layer 18, $\tau$=3, $\alpha$=1.1, $\beta$=0.1): **80.64**
>
> | Layer | 10 | 18 | 20 | 28 |
> | --- | --- | --- | --- | --- |
> | Score | 79.62 | 80.64 | 79.36 | 79.54 |
>
> | $\tau$ | 2.0 | 2.5 | 3.0 | 3.5 |
> | --- | --- | --- | --- | --- |
> | Score | 77.99 | 78.98 | 80.64 | 79.77 |
>
> | $\alpha$ | 1.05 | 1.075 | 1.1 | 1.125 |
> | --- | --- | --- | --- | --- |
> | Score | 80.31 | 80.14 | 80.64 | 78.85 |
>
> | $\beta$ | 0.2 | 0.15 | 0.1 | 0.05 | 0.0 |
> | --- | --- | --- | --- | --- | --- |
> | Score | 80.36 | 79.52 | 80.64 | 79.76 | 79.61 |
>
> Across all models (LLaVA-1.5, LLaVA-NeXT, Qwen2-VL), we observe similar trends regarding the optimal range for the layer, $\alpha$, $\beta$, and $\tau$ parameters—typically favoring mid-to-late transformer layers and slightly scaled values.
> We also note that our method is **training-free** and imposes **minimal additional computational cost**, making it practical and efficient to perform lightweight hyperparameter searches in real-world applications.

---

### Official Review · Reviewer_3wnt · 2025-03-13

**Overall Recommendation:** 3

**Summary:**

This work proposes a training-free method to enhance detailed image captioning with improved balance between precision and recall by re-calibrating the attention values in multimodal large language models (MLLMs). This work first analyzes the attention patterns in MLLMs and finds that 1) trivially enlarging the attention values leads to reduced diversity and lower recall in the captions, and 2) as more tokens are generated and the context becomes longer, the attention becomes more noisy and less focused on visual tokens. Then, this work proposes to select important visual tokens based on the attention dynamics and adjust the attention values accordingly. Experiments show improved overall quality of image captioning.

**Claims And Evidence:**

- Compared with baselines, the proposed method significantly improves the recall, but the precision is hurt. For example, as shown in Table 2, the precision is ~3% lower than PAI. Similarly, in Figure 6, the human evaluation suggests a lower precision compared with "Naive" (PAI). This may leads to concerns on hallucination and misleading information in the captions. Ideally, the method should be able to adjust its hyper-parameters to achieve a higher recall without hurting the precision.

**Essential References Not Discussed:**

No concerns.

**Experimental Designs Or Analyses:**

No concerns.

**Methods And Evaluation Criteria:**

- In addition to dense image captioning, it is suggested to include evaluation on other hallucination benchmarks and general VQA tasks, to ensure a more comprehensive comparison with baselines like PAI. Currently this work only tests CLAIR and CHAIR, which are a bit limited considering the scope of the related baselines.

- In the proposed method, there are a few hyper-parameters that need to be specified. Some are quite different across base MLLMs (see "Implementation Details" in Section 5.1). It might be challenging to select the hyper-parameters when applying the method to a new MLLM.

**Other Comments Or Suggestions:**

- It might be beneficial to include the actual inference cost comparison. As the method needs to monitor and manipulate attention computation in the model, the generation speed may be slowed.
- There are some typos to be fixed. For example, "Specifically, Specifically, " (page 4), "In Section Section 3" (page 4).

**Other Strengths And Weaknesses:**

No more concerns.

**Questions For Authors:**

No more questions.

**Relation To Broader Scientific Literature:**

This work proposes a new approach to dynamically augment the attention to visual tokens in MLLMs to produce better detailed image captioning. However, from the current evaluation, the improvement over existing methods seems not significant, and the usage cases might be limited.

**Theoretical Claims:**

This work does not include theoretical claims.

---

> ### Author Rebuttal · Authors · 2025-04-01
>
> **1. Concerns Regarding Performance Trade-off**
>
> > Compared with baselines, the proposed method significantly improves the recall, but the precision is hurt. For example, as shown in Table 2, the precision is ~3% lower than PAI. Similarly, in Figure 6, the human evaluation suggests a lower precision compared with "Naive" (PAI). This may lead to concerns on hallucination and misleading information in the captions. Ideally, the method should be able to adjust its hyperparameters to achieve a higher recall without hurting the precision.
> >
>
> We appreciate the reviewer’s concern regarding the potential trade-off between recall and precision in our method. However, we would like to clarify that, as shown in **Table 2**, our method achieves **improvements in both precision and recall** compared to the baseline. In contrast, other prior methods, including "Naive" (PAI), tend to **improve precision at the cost of recall**.
>
> In particular, while PAI demonstrates a higher precision than our method, it substantially reduces recall, which may not be acceptable for applications where **completeness of information is critical**. This is especially relevant in **real-world scenarios such as medical image reporting** or **automated content generation**, where omitting important visual elements may lead to misleading or insufficient descriptions. In such cases, **recall can be more crucial than precision**, as overly high precision may inadvertently filter out meaningful or necessary content.
>
> ---
> **2. Evaluation on other hallucination benchmark**
>
> Thank you for this valuable suggestion. While our primary focus is on **enhancing detailed image captioning**, we also performed additional experiments on the **POPE hallucination benchmark** [2],
>
> Inspired by the evaluation setup proposed in [1], where caption-based reasoning improves MLLM performance on general multimodal tasks, we adopt a similar strategy. Specifically, we first generate a caption for the image and then use it as part of the input to answer the question.
>
> We evaluated Qwen2-VL (7B) using three approaches: the baseline model, our proposed method, and naive attention scaling (PAI). For the naive method, we identified the optimal hyperparameter ($\alpha$ = 0.2) before evaluation.
>
> The table below summarizes the accuracy on the POPE benchmark:
>
> | Method | Accuracy (%) |
> | --- | --- |
> | Baseline | 82.01 |
> | Ours | 83.13 |
> | Naive Attn. Scaling | 81.45 |
>
> Our method shows an improvement over the baseline, while the naive attention scaling slightly reduces accuracy. Additionally, we evaluated the instruction-following behavior, measured as the proportion of responses in which the model correctly generates an output that includes both a caption and an answer.
>
> | Method | Instruction Following (%) |
> | --- | --- |
> | Ours | 92.72 |
> | Naive Attn. Scaling | 76.84 |
>
> These results indicate that naive attention scaling may reduce the model's sensitivity to the input prompt, whereas our method retains alignment with instruction while improving grounding accuracy.
>
> ---
> **3. Concerns Regarding the Choice of Hyperparameters**
>
> We appreciate the reviewer’s thoughtful concern regarding the generalizability of hyperparameter settings across different models. To address this, we conducted additional ablations on LLaVA-NeXT (7B) and Qwen2-VL (7B) using the DOCCI dataset. For detailed results, please refer to our response to **Reviewer 4gUZ**. Briefly, across all models (LLaVA-1.5, LLaVA-NeXT, Qwen2-VL), we observe similar trends regarding the optimal range for the layer, $\alpha$, $\beta$, and $\tau$ parameters—typically favoring mid-to-late transformer layers and slightly scaled values.
> We also note that our method is **training-free** and imposes **minimal additional computational cost**, making it practical and efficient to perform lightweight hyperparameter searches in real-world applications.
>
> ---
> **4. Computational Efficiency Comparison**
>
> Thank you for this valuable suggestion. A detailed comparison of runtime and memory overhead—including per-token generation latency and storage requirements—is provided in our response to **Reviewer JgJt**. To avoid redundancy, we kindly refer you to that section for a full breakdown. In brief, our method introduces only minimal overhead compared to the baseline, while remaining significantly more efficient than prior approaches.

---

> > ### Comment · Reviewer_3wnt · 2025-04-04
> >
> > The authors' response is greatly appreciated. Most of the previous concerns are addressed, so I will adjust my rating accordingly.
> >
> > Please include the details of additional evaluation and hyper-parameter ablation study in the revision.

---

> > > ### Author Response · Authors · 2025-04-06
> > >
> > > We thank the reviewer for the kind response. We are pleased to hear that the concerns have been addressed. As suggested, we will include the additional evaluation and hyper-parameter ablation study in the revision.

---

### Official Review · Reviewer_JgJt · 2025-03-14

**Overall Recommendation:** 4

**Summary:**

The paper introduces an ​adaptive attention enhancement mechanism aimed at improving the precision of image captioning while maintaining an acceptable recall rate. Specifically, the ​selective attention enhancement strategy seems powerful according to its significant improvement in the precision of long caption generation, effectively alleviating the ​hallucinations problem in MLLM.

**Claims And Evidence:**

The paper is well-motivated, with all insights grounded in ​empirical evidence, making the claims convincing. The paper proposes three main insights:
- ​Naive attention amplification reduces attention diversity, validated in Figure 2.
​- Noise increases with caption length, demonstrated in Figure 3.
- ​Visual focus weakens in long contexts, illustrated in Figure 5.
Through ​extensive experiments, the paper effectively proves the limitations of existing attention amplification methods and naturally derives the design of ​SPARC. The results also strongly support the proposed approach.

**Essential References Not Discussed:**

To my knowledge, the paper includes most related works.

**Experimental Designs Or Analyses:**

Yes. The method is evaluated by CLAIR measuring image-caption alignment and CHAIR measuring hallucination. The comparisons in Table 1 and 2 cover most of SOTA methods for attention strengthening The paper also evaluate their methods plugged into different MLLMs including LLaVA-1.5, LLaVA-Next, and Qwen, demonstrating the consistent effectiveness of SPARC.

**Methods And Evaluation Criteria:**

Yes. With EMA, the formulation of  Relative Activation Score seems plausible. The token selection and attention amplification methods based on the score are easy for implementation.

**Other Comments Or Suggestions:**

N/A

**Other Strengths And Weaknesses:**

## Strengths
- The observations validated by the paper may motivate many MLLM task, since hallucination in long-context scenarios is a generic problem in LLM. Besides, its attention patterns are also consistently investigated by the community.
## Weaknesses
- Since the paper involves additional attention amplification procedures,efficiency comparisons can be included, e.g., token generation speed.

**Questions For Authors:**

N/A

**Relation To Broader Scientific Literature:**

Precision dropping of MLLM in long caption generation remains a long-standing problem. Most of previous works focus on attention amplification but do not obtain satisfactory results. The paper reveals **the underlying causes supported by experiments** and propose a simple but effective method to address it, which is indeed meaningful.

**Theoretical Claims:**

The paper does not involve proofs.

---

> ### Author Rebuttal · Authors · 2025-04-01
>
> We sincerely thank the reviewer for taking the time and effort to evaluate our paper. We truly appreciate your insightful comments. Please find our detailed response to your comment below. If you have any further feedback, we would be grateful to hear it.
>
> **1. Efficiency Comparisons**
>
> > Since the paper involves additional attention amplification procedures, efficiency comparisons can be included. e.g., token generation speed.
> >
>
> Thank you for your valuable suggestion. While our method involves attention amplification procedures, these introduce **only minimal overhead** compared to the original decoding process. In contrast, many previously proposed approaches require **additional decoding passes**, resulting in significant computational cost.
>
> To provide a clearer comparison, we measured the **token generation time** (i.e., generation time per output token) across various methods. The following table presents the average generation time per token (in milliseconds) using an **RTX8000 GPU**:
>
> | Method | Token Generation Time (ms/token) |
> | --- | --- |
> | Baseline | 30.37 ± 0.73 |
> | Ours | 31.21 ± 0.61 |
> | Volcano | 109.98 ± 17.71 |
> | PAI | 57.75 ± 0.86 |
> | VCD | 59.44 ± 0.84 |
> | OPERA | 322.28 ± 118.26 |
>
> As shown, our method performs similarly to the baseline, whereas other methods introduce 2x to 10x slower generation speeds. This demonstrates that our approach achieves **efficient generation with minimal computational overhead**.
>
> Regarding memory usage: our method only requires storing the **head-wise averaged attention scores for image tokens** at each layer from the previous decoding step. For example, in the case of LLaVA-1.5 (7B), this amounts to:
>
> 32 (layers) × 576 (image tokens) × 2 bytes (float16) < 40 KB
>
> This overhead is negligible in modern hardware setups.

---

### Official Review · Reviewer_kRcu · 2025-03-14

**Overall Recommendation:** 4

**Summary:**

The authors study the effect of attention variability spatially and temporally and its impact on detailed image captioning with Visual Language Models (VLMs). The authors provide a detailed analysis of methods that tackle attention leaking from the image into the text as the caption grows, and they find that simply increasing the attention on the image makes the captioning focus on only a few objects and noise. Therefore, existing methods that improve the precision of captions do so at the cost of the recall. To mitigate this, they propose a new attention rescaling framework that highlights tokens whose attention scores frequently vary with the captioning process. They show that this approach can improve the alignment with reference captions in relevant datasets with three state-of-the-art VLMs.

## Update after rebuttal
The authors have satisfied all my requests, and I hope the new results and discussions are reflected in the final version. Therefore, I update my score to "Accept".

**Claims And Evidence:**

The authors show a detailed analysis of naive attention scaling and a proposed method to numerically improves image captioning, based on the diversity of attention scores throughout the captioning generation. Attention diversity caused by their method is effectively shown in Fig 11c, but it is unclear whether this diversity is allocated to more or less “noisy” scores. The only example is Figure 2, which indeed shows a significant amount of “noisy” scores with the proposed method.


The authors base a significant part of their analysis on the attention scores of a querying token from the caption to the image tokens at a mid-to-late layer of the model. I agree with the implied statement that tokens preserve significant information relating to their initial embedding, but multiple layers of attention should also lead to a significant mixing of information. Therefore, asserting that a particular attention score is “noisy” in this setting needs more substantial evidence. If feasible, a saliency map of the image patches w.r.t. the attention scores of that layer could make this claim more solid. Besides, it is unclear to me that such “noisy” attention scores are inherently harmful as they have been shown to be used for computation by ViTs [1].

Furthermore, the authors claim that this attention diversity correlates with the model not focusing on only a few relevant objects, therefore improving recall. This seems clearly supported by their improved recall scores.


[1] Darcet et al. Vision Transformers Need Registers. ICLR’24.

**Essential References Not Discussed:**

The assumption that high attention scores in mid-to-late layers to tokens related to background patches, is harmful or “noisy” is not empirically supported. No relevant literature is mentioned on that aspect. In contrast, relevant works [1,6] show that large attention scores and/or activations are a key component of ViTs, and can even be employed for better performance.

[6] Sun et al. Massive Activations in Large Language Models. COLM’24.

**Experimental Designs Or Analyses:**

Overall, the authors chose two datasets and two reasonable and comprehensive metrics and showed their approach in different models. I would appreciate an analysis of the “noisy” scores for their approach and naive attention scaling.
I would also appreciate ablations like those shown in Tables 6-9 for all models, ideally in all datasets. Even if fewer parameters are tried (e.g., only three values per hyperparameter), it would be good to see how sensitive these are across models since, e.g., the final result for Qwen-2-VL uses a different layer.

**Methods And Evaluation Criteria:**

After a detailed analysis of the limitations of current methods, the authors propose a reasonable method to mitigate the observed shortcomings. The metrics used to evaluate this method, namely object-matched and GPT-4o scored matching between gold captions and generated captions, make sense.

**Other Comments Or Suggestions:**

Is Figure 5 weighting for the decreasing share of image tokens w.r.t. total tokens? If not, I think that is not very useful. It seems intuitive that as captioning goes on and the image tokens aren’t 100% of the previous tokens, the model has to devote some attention to the text it has already been output. I am not saying they should be weighted the same (clearly from previous work, they are not), but this plot would assume that text tokens need not be attended to. A more reasonable plot would weigh the attention score by the share of total tokens from each type.
Typos:
L204: “Specifically, Specifically, “
L350: “We compar our method”

**Other Strengths And Weaknesses:**

Main strengths:
- Detailed analysis of naive scaling of attention for image captioning, and proof that this correlates with recall in image captioning
- New training-free method to improve recall and precision in image captioning.
- Well written paper.
Summary of weaknesses:
- No analysis of “noisy” attention scores. If feasible, use saliency map w.r.t. attention scores instead of attention scores directly; in such a deep layer, these might not correspond only to the original patch.
- Fig 5 does not account for a share of total tokens.
- No ablations like tables 6-9 for other models & other datasets already shown in the paper. Unclear how difficult it’d be to tune this framework in a new model.
- The resulting variation of attention scores applying their method is currently in the appendix but seems relevant enough to be in the main paper.

**Questions For Authors:**

L372: “To ensure a robust evaluation, we randomly sample 500 instances and repeated the evaluation five times.” Does the “repeating” here come from 500 sampling subsets or for generating captions with non-determinism (temperature different than 0)? This could be made clearer, and if the reason is the latter, more details on the generation parameters should be provided.

**Relation To Broader Scientific Literature:**

The authors discuss prior work on improving captioning, specifically precision, by mitigating hallucinations. Recent work in this direction [2,3,4] tries to focus the textual generation process more on the visual input. Such past works show the benefits not only for image captioning but for other tasks such as visual question answering. However, the authors of this work show that temporally varying attention is key for higher recall and that simply scaling attention leads to higher precision at the cost of lower recall. In this sense, these findings could also be applied to other image-text tasks.

It is, however unclear how other techniques [5] that do not rely on increasing attention scores for improving visual attention in VLMs compare to the proposed method. [5] shows they can slightly improve recall while making larger improvement on CHAIR specific metrics.

[2] Huo et al. Self-Introspective Decoding: Alleviating Hallucinations for Large Vision-Language Models. ICLR’25.
[3] Liu et al. Paying more attention to image: A training-free method for alleviating hallucination in lvlms. ECCV’24.
[4] Li et al. Mitigating Hallucination for Large Vision Language Model by Inter-Modality Correlation Calibration Decoding. arXiv’25.
[5] Xing et al. Mitigating Object Hallucination via Concentric Causal Attention. NeurIPS’24.

**Theoretical Claims:**

No theoretical claims.

---

> ### Author Rebuttal · Authors · 2025-04-01
>
> We sincerely thank the reviewer for the thoughtful feedback and insightful question. Your comments greatly helped us refine and clarify the paper. Please find our detailed response below—we’re happy to address any further concerns.
>
> **1. Analysis of the “noisy” scores**
>
> We understand the reviewer’s concern regarding whether the increased attention diversity in our method may be attributed to “noisy” scores. To investigate this, we conducted an analysis of attention “noisiness” by measuring the proportion of attention assigned to **semantically relevant image regions** during decoding, using 5,000 randomly sampled images from the MSCOCO 2014 validation set.
>
> Specifically, for each generated token corresponding to a ground truth object word in the caption, we computed the **proportion of total attention allocated to image tokens within the region of that object**, relative to the total attention across all image tokens. These object regions were identified using binary masks obtained from an open-vocabulary segmentation model, Grounded SAM2 [1], applied to each object in the image.
>
> The table below compares the average attention allocated to relevant image regions across three methods: Baseline, Ours, and Naive Attention Scaling. A higher proportion indicates that the model is focusing more accurately on the relevant visual regions, suggesting lower attention noise.
>
> | Method | Attention on Relevant Image Regions (%)|
> | --- | --- |
> | Baseline | 17.85 |
> | Ours | 19.17 |
> | Naive Attn. Scaling | 15.50 |
>
> These results indicate that our method assigns more focused attention to semantically meaningful image areas compared to the baseline and naive attention scaling, suggesting that our method results in *less noisy* visual attention.
>
> Furthermore, in Fig. 13(a), we report the **caption similarity score**, which measures the semantic similarity between sentences within a generated caption. Captions generated by our method show greater diversity in sentence content, reinforcing that the diversity is not due to randomness or noise in attention, but rather stems from meaningful differentiation in visual grounding and language generation.
>
> [1] Ren et al. Grounded SAM 2: Ground and Track Anything in Videos. IDEA-Research, GitHub, 2024.
>
> ---
> **2. More Ablations for Parameters**
>
> We appreciate the reviewer’s suggestion regarding additional ablation studies on different models and datasets. To address this point, we conducted additional ablation studies on LLaVA-NeXT (7B) and Qwen2-VL (7B) using the DOCCI dataset, analogous to Tables 6–9 in the paper. To avoid redundancy, we kindly refer you to our response to **Reviewer 4gUZ**, where detailed results and observations are provided. In brief, we observed consistent trends across models, with performance generally favoring mid-to-late layers and modest parameter scaling. Since our method is training-free and efficient, such tuning remains practical.
>
> **3. Clarification Regarding “Noisy” Attention Scores**
>
> We sincerely apologize for any confusion caused by our use of the term “noisy” to describe certain attention scores. As discussed in [2], background tokens can carry important global context and are often beneficial in vision transformers.
>
> Our intent was not to imply that such attention is inherently harmful. Rather, our goal is to **enhance attention to semantically relevant visual regions** during decoding—not to suppress background tokens altogether. In fact, we agree that background tokens can carry useful information, and our method does not explicitly penalize them.
>
> However, we found that naive attention scaling may unintentionally amplify attention to globally dominant tokens (including background regions), which can **overwhelm local, task-relevant signals**. This may explain why captions generated with naive attention scaling often exhibit high precision but reduced recall—they tend to overemphasize prominent visual cues while neglecting finer details.
>
> We sincerely thank the reviewer for pointing this out and will revise the manuscript to replace “noisy” with more accurate wording.
>
> [2] Darcet et al. Vision Transformers Need Registers. ICLR’24.
>
> **4. Attention weight trends for text and image tokens as context length increases**
>
> > Fig. 5 does not account for the share of total tokens.
> >
>
> Thank you for the insightful comment. As noted, Fig. 5 did not normalize for the total number of tokens. To address this, we analyzed the average attention per token and found that attention to image tokens decreases disproportionately faster than to text tokens as context length increases.
>
> **5. Clarification on the meaning of “repeating” in evaluation**
>
> Thank you for the question. The “repeating” refers to resampling 500 instances from the MSCOCO validation set five times, following prior work [3]. Generation was deterministic with temperature set to 0.
>
> Liu et al. Paying more attention to image: A training-free method for alleviating hallucination in lvlms. ECCV’24. [3]

---

> > ### Comment · Reviewer_kRcu · 2025-04-02
> >
> > I thank the authors for additional experiments and clarifications. I would like to raise several points:
> >
> > **1. Analysis of the “noisy” scores**
> > I initially cited as a weakness “No analysis of “noisy” attention scores. If feasible, use saliency map w.r.t. attention scores instead of attention scores directly; in such a deep layer, these might not correspond only to the original patch.” It seems this later suggestion has not been considered, although I believe it is very relevant.
> >
> > Furthermore, the presented analysis does not address the doubt that more attention diversity does not come from “noisy” tokens. This new experiment measures attention from text tokens of objects to relevant image patches. However, the attention diversity is measured from all text tokens, including those not referring to objects. Therefore, although this experiment is insightful and shows that attention with the proposed method aligns better with relevant patches, it doesn’t help in understanding the nature of the improved attention diversity (which authors deem key to better captioning recall) or true faithfulness to input (since attention scores of tokens at a deep layer might not relate to the original patch).
> >
> > **2. More Ablations for Parameters**
> > This ablation is helpful. Unfortunately, the optimal setting for each model is not ablated for all other models, e.g. the optimal setting for LLaVA-NeXT-7B is not shown for Qwen-2-VL. Therefore, the impact of using the existing hyperparameters on a new model is unclear. Since some of these hyperparameter settings (e.g. if one would use layer 20 as optimal for LLaVA-NeXT-7B in Qwen-2-VL) would offer little benefit over the baseline, it should be made clear in the paper that tuning these parameters is quite essential.
> >
> > **3. Clarification Regarding “Noisy” Attention Scores**
> > I thank the authors for the clarification. I believe the re-framing of “noisy” tokens to tokens with global context could help clarity. However, as I mentioned in my original review and reiterated in point 1, I believe an analysis beyond attention scores, for instance, using saliency scores, would have shown more faithfully the relationship to the input regions.
> >
> > **4. Attention weight trends for text and image tokens as context length increases**
> > Thank you for the corrected plot.
> >
> > **5. Clarification on the meaning of “repeating” in evaluation**
> > Thank you for the clarification. It is still unclear why one would not sample 5x500 samples without replacement from the MSCOCO validation set once. This would be more statistically sound at the same cost, as I understand it.

---

> > > ### Author Response · Authors · 2025-04-04
> > >
> > > Thank you very much for taking the time to read our rebuttal and for engaging in this discussion. We truly appreciate your thoughtful feedback and would like to address your follow-up comments below. Please feel free to let us know if you have any further questions or suggestions.
> > >
> > > ---
> > >
> > > **1. Concerns Regarding the Use of Attention Scores for Analysis**
> > >
> > > > "If feasible, use saliency map w.r.t. attention scores instead of attention scores directly; in such a deep layer, these might not correspond only to the original patch."
> > > >
> > >
> > > Thank you for the insightful suggestion. Following your recommendation that employing saliency maps with respect to attention scores — rather than relying solely on attention scores — could provide a more reliable basis for our analysis, we conducted additional experiments.
> > >
> > > Specifically, we computed and visualized a **saliency map by weighting attention scores at the given layer with their gradients with respect to the model’s output**. Our implementation follows the gradient-weighted attention approach introduced in [1].
> > >
> > > In the figure linked [here](https://anonymous.4open.science/r/images-EBFF/saliency_map.pdf), we compare the results of the original analysis in Figure 4 of our paper with the new results using the saliency maps. We observe that the saliency-based analysis exhibits similar trends: as the context length increases during caption generation, the saliency map becomes increasingly noisy. This supports our original interpretation. Moreover, recent work such as [1] has shown that the attention patterns of MLLMs do align with relevant image patches, especially in tasks like visual question answering.
> > >
> > > [1] Zhang et al., MLLMs Know Where to Look: Training-free Perception of Small Visual Details with Multimodal LLMs, ICLR’ 2025.
> > >
> > > > "Furthermore, the presented analysis does not address the doubt that more attention diversity does not come from 'noisy' tokens. This new experiment measures attention from text tokens of objects to relevant image patches. However, the attention diversity is measured from all text tokens, including those not referring to objects."
> > > >
> > >
> > > We are grateful for your thoughtful observation. To address this concern, we conducted an additional experiment in which we **measured attention diversity using only the patches corresponding to actual objects** in the image, rather than across the entire image. This was done to ensure that the diversity values are not influenced by background noise.
> > >
> > > In particular, for each image in the MSCOCO 2014 validation set (5,000 images), we identified the **foreground regions** by aggregating the ground-truth segmentation masks of all annotated objects. During caption generation, we collected the attention scores of image tokens and retained only those corresponding to the foreground patches. We then computed attention diversity using the same methodology as in Figure 11 of the paper.
> > >
> > > The figure linked [here](https://anonymous.4open.science/r/images-EBFF/diversity.pdf) plots attention diversity across foreground regions for the baseline, naive scaling, and our method. As with Figure 11, our method exhibits higher attention diversity than naive scaling, even when noise from background regions is excluded.
> > >
> > > We hope this addresses your concern and strengthens the interpretation of our results. Thank you again for your valuable feedback.
> > >
> > > ---
> > >
> > > **2. Concerns Regarding Hyperparameter Settings**
> > >
> > > Thank you for highlighting the importance of hyperparameter tuning. As you pointed out, identifying optimal parameter settings is indeed necessary to achieve the best possible performance for each model. We will make it clear in the revised version of the paper that tuning these hyperparameters is essential when applying our method to different architectures.
> > >
> > > That said, as previously mentioned, since our method is training-free and computationally lightweight, we believe that identifying optimal hyperparameters for new models is relatively straightforward and does not pose a significant barrier in practical use.
> > >
> > > ---
> > >
> > > **3. Regarding Repeated Sampling**
> > >
> > > Thank you for your helpful comment. Following your suggestion, we re-evaluated the experiments reported in Table 2 of the paper using a single set of **2,000 samples drawn without replacement** from the MSCOCO validation set. The results are shown in the table below:
> > >
> > > | Method | Precision | Recall | F1 |
> > > | --- | --- | --- | --- |
> > > | Baseline | 84.72 | 79.55 | 82.05 |
> > > | OPERA | 84.76 | 79.30 | 81.94 |
> > > | VCD | 83.55 | 77.79 | 80.57 |
> > > | VOLCANO | 87.58 | 77.91 | 82.46 |
> > > | PAI | **90.94** | 72.31 | 80.56 |
> > > | Ours | 87.54 | **80.16** | **83.69** |
> > >
> > > The reason we originally used 5×500 samples was to align with the setup used in prior work, which typically evaluated on 500 samples.

---

### Decision · Program_Chairs · 2025-05-01

**Decision:**

Accept (poster)

**Comment:**

The authors did a good rebuttal, with the final rating as 2 weak accept and 2 accept. The AC has checked the review, the discussion, the manuscript, and the rebuttal, and agrees with this assessment.